# Optimal Regret Bounds via Low-Rank Structured Variation in Non-Stationary Reinforcement Learning

**Tuan Dam**
Hanoi University of Science and Technology, Hanoi, Vietnam
tuandq@soict.hust.edu.vn

## Abstract

We study reinforcement learning in non-stationary communicating MDPs whose transition drift admits a low-rank plus sparse structure. We propose **SVUCRL** (Structured Variation UCRL) and prove the dynamic-regret bound

$$\tilde{\mathcal{O}}\Big(\sqrt{SAT} + D_{\max}S\sqrt{AT} + L_{\max}B_r + D_{\max}L_{\max}B_p + D_{\max}S\sqrt{AB_p} + D_{\max}\delta_B B_p + D_{\max}\sqrt{KT}\Big),$$

(up to the additional planning-tolerance term $\sum_{t=1}^{T} \varepsilon_{\tau(m(t))}$).

where $S$ is the number of states, $A$ the number of actions, $T$ the horizon, $D_{\max}$ the MDP diameter, $B_r/B_p$ the total reward/transition variation budgets, and $K \ll SA$ the rank of the structured drift, $L_{\max}$ is the maximum episole length. The first two terms are the statistical price of learning in stationary problems. The structure-dependent non-stationarity contribution appears through $D_{\max}\sqrt{KT}$ (low-rank drift) and $D_{\max}\delta_B B_p$ (sparse shocks), which scale with $\sqrt{K}$ rather than $\sqrt{SA}$ when drift is low-rank. This matches the $\sqrt{T}$ rate (up to logs) and improves on prior $T^{3/4}$-type guarantees. SVUCRL combines: (i) online low-rank tracking with explicit Frobenius guarantees, (ii) incremental RPCA to separate structured drift from sparse shocks, (iii) adaptive confidence widening via a bias-corrected local-variation estimator, and (iv) factor forecasting with an optimal shrinkage center.

## 1 Introduction

Reinforcement learning (RL) algorithms have achieved remarkable success in stationary environments with fixed reward distributions and state transition dynamics. However, many real-world applications involve non-stationary environments where dynamics evolve due to changing user preferences, environmental conditions, or system parameters. This non-stationarity poses significant challenges for traditional RL approaches that assume fixed environment dynamics.

Non-stationary RL faces environments whose reward and transition laws evolve in complex ways. Standard approaches use sliding-window techniques that focus on recent observations while discarding older data. Algorithms such as SWUCRL2–CW [6] widen confidence sets to cover temporal drift, providing theoretical guarantees at the cost of higher regret.

These existing approaches have a significant limitation: they use uniform widening parameters that ignore the *structure* of environmental evolution. In many real systems, however, drift exhibits exploitable patterns: it often lives in low-dimensional subspaces ($K \ll SA$) where only a few underlying factors drive changes; the changes follow smooth trajectories enabling short-term forecasting; and many environments exhibit structured evolution with occasional sparse shocks affecting only small subsets of state-action pairs.

39th Conference on Neural Information Processing Systems (NeurIPS 2025).

This paper leverages these observations to develop SVUCRL (Structured Variation UCRL), which combines matrix factorization, robust statistics, and time-series analysis to achieve improved regret bounds and computational efficiency.

Our contributions:

1. **Structured variation model and $\delta_B$.** We model drift as low-rank plus a sparse component whose $\ell_1$-mass consumes at most a $\delta_B$-fraction of the transition-variation budget $B_p$. This separates the learnable, shared dynamics from idiosyncratic shocks (Assumption 1).

2. **Provable low-rank tracking.** A power–Frobenius inequality and randomized SVD with power iterations control the streaming approximation error with explicit constants (Lemmas 1, 2).

3. **Shock isolation (PCP).** We (conceptually) use the convex PCP/RPCA program (1) to separate structured low-rank drift from sparse shocks, with an exact recovery guarantee under standard incoherence and random-support conditions (Proposition 1). Algorithm 2 is used as a fast incremental subspace update to warm-start/accelerate PCP computations.

4. **Forecast–shrinkage center.** Forecasted factors define a low-variance center that is combined with empirical transitions via a James–Stein weight; the data-driven weight is asymptotically optimal (Theorem 1).

5. **Main regret bound.** Summing per-step bounds under the above controls yields the three-term dynamic-regret guarantee, matching $\sqrt{T}$ rates (Theorem 2).

The resulting algorithm, SVUCRL, enjoys the regret bound of Theorem 2 and is computationally $\mathcal{O}(TSA(SK + S)\log T)$.

**Notation** $S = |\mathcal{S}|$ denotes the size of the state space, $A$ is the average number of actions, $D_{\max}$ the diameter of the MDP, $B_r, B_p$ the reward/transition variation budgets, and $\|\cdot\|_{1,\infty}$ the maximum row $\ell_1$ norm. Throughout $\widetilde{\mathcal{O}}$ hides $\mathrm{polylog}(T, S, A)$ factors.

**Related Work** Non-stationary reinforcement learning has been studied under various modeling assumptions. The sliding window approach has been explored extensively [6, 7], with algorithms that discard data outside a recent window. Change-point detection methods [16] attempt to identify significant shifts in environment dynamics. Bandit-based approaches [4, 17] use various weighting schemes to prioritize recent observations. Our work is most closely related to the confidence widening approach in SWUCRL2-CW [6], but we significantly improve upon it by exploiting structure in the variation. Our structured variation model bears some similarity to factored MDPs [12, 15], but we focus on the structure of *changes* rather than the structure of the MDP itself. The matrix decomposition techniques we employ relate to robust PCA [5] and online matrix factorization [13], but we adapt these methods to the specific challenges of sequential decision-making under non-stationarity. Our adaptive confidence widening connects to adaptive concentration inequalities in statistics [9, 10]. Beyond classical factored MDPs [12, 15], recent work explores low-rank structure for sample-efficient control and representation learning, e.g., low-rank MDPs with continuous actions [3] and model-based methods that exploit low-rank structure [1]. Our setting differs by allowing *time-varying* dynamics with a low-rank drift plus sparse shocks, and our analysis quantifies how exploiting this structure improves dynamic-regret rates.

Our $\tilde{O}(\sqrt{T})$ dependence does not contradict known lower bounds in the *unstructured* non-stationary setting: for communicating MDPs, Mao et al. [14] show any algorithm suffers at least $\Omega\big((B_r+B_p)^{1/3}T^{2/3}\big)$ when no structure is assumed. By contrast, our improvement *leverages* low-rank drift and sparse shocks. Unless otherwise stated, all comparisons in this paper are made under Assumption 1 (low-rank drift plus sparse shocks); in the same regime, SWUCRL2-CW [6] is the most relevant baseline.

## 2 Problem set-up

We study reinforcement learning in non-stationary environments, formalized as a sequence $\big(\mathcal{S}, \mathcal{A}, p_t, r_t\big)_{t=1}^{T}$ of communicating Markov Decision Processes (MDPs) with diameter at most $D_{\max}$.

**MDP Sequence** Each MDP $M_t$ in the sequence shares the same state space $\mathcal{S}$ and action space $\mathcal{A}$ but has potentially different transition dynamics $p_t$ and reward functions $r_t$ at each time step $t$. The transition function $p_t(s'|s, a)$ specifies the probability of transitioning to state $s'$ when taking action $a$ in state $s$ at time $t$. Similarly, the reward function $r_t(s, a)$ represents the expected reward for taking action $a$ in state $s$ at time $t$.

**Communicating MDPs** We assume that each MDP in the sequence is communicating, meaning that for any pair of states $s, s' \in \mathcal{S}$, there exists a policy that reaches $s'$ from $s$ with non-zero probability. The diameter $D_{\max}$ quantifies the worst-case expected time to navigate between any two states, providing a measure of the connectivity of the MDP.

**Variation Budgets** To quantify the degree of non-stationarity, we define variation budgets for both rewards and transitions: $B_r = \sum_t \max_{s,a} |r_{t+1}(s, a) - r_t(s, a)|$, $B_p = \sum_t \max_{s,a} \|p_{t+1}(\cdot|s, a) - p_t(\cdot|s, a)\|_1$. The reward variation budget $B_r$ measures the cumulative maximum change in rewards across all state-action pairs, while the transition variation budget $B_p$ captures the cumulative maximum change in transition probabilities measured in $\ell_1$ norm. These budgets provide a formal way to bound the total amount of non-stationarity in the environment.

**Learning Protocol** The learning process proceeds as follows: at each time step $t$, the agent observes the current state $s_t$, selects an action $a_t$ based on its policy, and the environment generates a reward $r_t(s_t, a_t)$ and transitions to the next state $s_{t+1}$ according to $p_t(\cdot|s_t, a_t)$. The agent then updates its policy based on the observation $(s_t, a_t, r_t, s_{t+1})$.

**Dynamic regret** The performance of a learning algorithm is measured by its dynamic regret, defined as:

$$\text{DynReg}_T = \sum_{t=1}^{T} \left( \rho_t^* - r_t(s_t, a_t) \right)$$

where $\rho_t^*$ is the optimal average reward for MDP $t$ (achievable by an oracle that knows the dynamics of MDP $t$ in advance) and define as

$$\rho_t^* := \sup_\pi \lim_{T \to \infty} \frac{1}{T} \mathbb{E}_\pi^{M_t} \left[ \sum_{i=0}^{T-1} r_t(s_i, a_i) \right],$$

where $a_i \sim \pi(\cdot \mid s_i)$ and $s_{i+1} \sim p_t(\cdot \mid s_i, a_i)$. The dynamic regret measures the cumulative difference between the reward obtained by the algorithm and the reward that could have been obtained by an optimal policy for each MDP in the sequence. This is a more challenging metric than the static regret often used in stationary environments, as it requires the algorithm to track the changing optimal policy over time.

**Challenges** Non-stationary RL presents several key challenges. The exploration-exploitation tradeoff requires the agent to balance exploring to learn the changing dynamics with exploiting current knowledge to maximize reward. The agent must also demonstrate adaptivity by adapting quickly to changes in the environment without discarding too much relevant historical data. Additionally, computational efficiency is crucial as processing the continuous stream of observations requires efficient algorithms, especially for large state and action spaces.

Our approach addresses these challenges by exploiting structure in the environmental changes, allowing for more efficient learning and better adaptation to the evolving dynamics.

## 3 Structured variation model

The core insight of our approach is that changes in the environment dynamics often exhibit structure that can be exploited for more efficient learning. We formalize this intuition in our structured variation model. Most of the *change* in the dynamics from step $t$ to $t+1$ can be explained by a few latent drivers, plus occasional localized shocks.

**Assumption 1** (Low-rank drift). For each $t$, the transition change $\Delta P_t := P_{t+1} - P_t \in \mathbb{R}^{SA \times S}$ admits the decomposition

$$\Delta P_t = \sum_{k=1}^{K} u_k(t) \underbrace{v_k}_{\in \mathbb{R}^{SA}} \underbrace{w_k^\top}_{\in \mathbb{R}^S} + \epsilon_t, \qquad \sum_t \max_{s,a} \|\epsilon_t(s, a, \cdot)\|_1 \leq \delta_B B_p,$$

with per-factor bounds $\|w_k\|_1 \leq 1$ and $|v_k(s, a)| \leq 1$ for all $(s, a)$.

In this assumption:

- $u_k(t)$ is the *time weight* of factor $k$ at step $t$.
- $v_k$ is a pattern over state–action rows $(s,a)$ saying *which* parts of the MDP are affected by factor $k$.
- $w_k$ is a pattern over next states $s'$ saying *how* probability mass is reallocated when factor $k$ acts.
- $\epsilon_t$ is a *sparse shock* capturing rare, localized, hard-to-predict changes.

**Why the constraints matter.** The simple bounds $|v_k(s,a)| \leq 1$ and $\|w_k\|_1 \leq 1$ are a convenient *scaling convention*: they push the overall magnitude of each factor into $u_k(t)$. This makes the per-step change interpretable and ensures that $\max_{s,a} \| \sum_k u_k(t) v_k(s,a) w_k \|_1$ is directly controlled by $\sum_k |u_k(t)|$. The budget $\sum_t \max_{s,a} \|\epsilon_t(s,a,\cdot)\|_1 \leq \delta_B B_p$ says that shocks consume at most a $\delta_B$–fraction of the total transition variation $B_p$, so most drift is structured.

**What the model buys you.** Low rank means *shared structure across rows*: many $(s,a)$-rows move in a correlated way (via the same $w_k$ pattern), at amplitudes set by $v_k(s,a)$, and with time profiles $u_k(t)$. Under this structural assumption, SVUCRL does not need to treat the non-stationary transition component as an arbitrary $SA$-dimensional drift process; instead, it can summarize the time variation through a $K$-dimensional factorization. In the regret bound (Theorem 2), this manifests through a *structured approximation* contribution $\widetilde{\mathcal{O}}(D_{\max}\sqrt{KT})$ and a sparse-shock residual $D_{\max}\delta_B B_p$. When $K \ll SA$ and shocks are limited (small $\delta_B$), these structure-dependent terms can be substantially smaller than what is typically incurred under worst-case, unstructured drift models. The remaining non-stationarity cost is captured by the within-episode drift terms $L_{\max}B_r + D_{\max}L_{\max}B_p$ (and the widening term $\widetilde{\mathcal{O}}(D_{\max}S\sqrt{AB_p})$), which reflect how changes accumulate over the episode length.

This model is motivated by several observations about real-world systems. In traffic and control settings, a weather factor can shift many transitions in the same direction (e.g., wet roads), where $v_k$ highlights the affected links, $w_k$ encodes where mass moves (such as slower lanes), and $u_k(t)$ follows the storm's intensity. Similarly, in recommendation systems, a global popularity wave reweights next-state preferences for many user contexts in tandem, with occasional item-specific shocks handled by $\epsilon_t$.

**How it relates to the variation budgets.** Recall $B_p = \sum_t \max_{s,a} \|P_{t+1}(\cdot|s,a) - P_t(\cdot|s,a)\|_1$. Under Assumption 1, the *structured* part of each row change is $\sum_k u_k(t) v_k(s,a) w_k$, whose $\ell_1$-size per row is at most $\sum_k |u_k(t)|$ by the bounds on $v_k, w_k$. Hence the same few coefficients $u_k(t)$ concurrently govern the row-wise maxima that enter $B_p$, this is the leverage SVUCRL exploits.

**Sanity checks.**

- *Stationary case:* if the environment is stationary, then $\Delta P_t \equiv 0$ and one can take $K = 0$, $\epsilon_t \equiv 0$.

- *Many weak drivers:* as $K$ grows or $\delta_B$ increases, the advantage diminishes smoothly; the model reduces to general variation when $K$ approaches $SA$ or shocks dominate.

**No need to know $K$ a priori.** SVUCRL *estimates* an effective rank online (via randomized SVD with oversampling and power iterations), and updates it as the spectrum evolves; the algorithm and guarantees do not require the true $K$ as input.

**What this model is *not*.** We do *not* assume the MDP itself is factored; only the *drift* $\Delta P_t$ is approximately low rank plus sparse. This distinction lets us capture global but compact changes even when the underlying $P_t$ has no simple structure.

## 4 Online low-rank approximation

A key component of our approach is efficiently tracking the low-rank structure of environmental changes as they evolve over time. This section develops the theoretical and algorithmic foundations for this tracking process.

To represent the changes in transition dynamics, we flatten the state-action pairs $(s,a)$ into rows and the next-state indices into columns, forming matrices. We denote by $\mathbf{X}_t = [\Delta P_{t-W+1}, \ldots, \Delta P_t] \in \mathbb{R}^{SA \times WS}$ the matrix containing the last $W$ changes in transition probabilities.

---
**Algorithm 1** Randomised SVD with power iterations

---
**Require:** matrix $\mathbf{X}$, target rank $\widehat{K}$, oversampling $s$, iters $q$
1:  $\Omega \leftarrow \mathcal{N}(0,1)^{WS \times (\widehat{K}+s)}$
2:  $\mathbf{Y}_0 \leftarrow \mathbf{X}\Omega$
3:  **for** $j = 1$ to $q$ **do**
4:      $\mathbf{Y}_j \leftarrow \mathbf{X}(\mathbf{X}^\top \mathbf{Y}_{j-1})$
5:  **end for**
6:  $\mathbf{Q} \leftarrow \mathrm{qr}(\mathbf{Y}_q)$
7:  $\mathbf{B} \leftarrow \mathbf{Q}^\top \mathbf{X}$
8:  $\mathbf{U}_B, \Sigma, \mathbf{V} \leftarrow \mathrm{svd}(\mathbf{B})$
9:  $\mathbf{U} \leftarrow \mathbf{Q}\mathbf{U}_B$
10: **return** $(\mathbf{U}_{[:,1:\widehat{K}]}, \Sigma_{1:\widehat{K}}, \mathbf{V}_{[:,1:\widehat{K}]})$

---

## 4.1 A Frobenius power-iteration bound

We begin by establishing a theoretical result that relates the Frobenius norm error of a low-rank approximation to that of a power-iterated version of the matrix. This result is crucial for the theoretical guarantees of our randomized SVD algorithm.

**Lemma 1** (Power–Frobenius). Let $X = U\Sigma V^\top$ be the singular value decomposition of a (real) matrix $X$, and let

$$B := (XX^\top)^q X \qquad \text{for an integer } q \geq 0.$$

For any rank-$\widehat{K}$ orthogonal projector $P$ whose range is contained in the column space of $X$ (i.e., $\mathrm{range}(P) \subseteq \mathrm{range}(X)$), define $m := \mathrm{rank}(X) - \widehat{K}$ (the tail dimension). Then

$$\left\|(I-P)X\right\|_F \leq m^{\frac{q}{2q+1}} \left\|(I-P)B\right\|_F^{\frac{1}{2q+1}}.$$

In particular, if $\mathrm{rank}(X) \leq 2\widehat{K} + 1$ (so $m \leq \widehat{K} + 1$), then

$$\left\|(I-P)X\right\|_F \leq (\widehat{K}+1)^{\frac{q}{2q+1}} \left\|(I-P)B\right\|_F^{\frac{1}{2q+1}}.$$

This lemma establishes that the error of a rank-$\widehat{K}$ approximation of $\mathbf{X}$ is related to the error of approximating the power-iterated matrix $\mathbf{B}$. Intuitively, power iteration amplifies the gap between the top $\widehat{K}$ singular values and the remaining ones, making it easier to identify the dominant subspace. The lemma quantifies this relationship with explicit constants, showing that the error decreases exponentially with the number of power iterations $q$.

## 4.2 Randomised SVD with explicit constants

Building on the theoretical foundation of the previous section, we now present our randomized SVD algorithm with explicit error guarantees.

**Lemma 2** (Online low–rank estimator). Run Algorithm 1 with oversampling $s \geq 3$ and $q \geq 0$ power iterations, and let $\widehat{K} = \widehat{K}_t$. Define the tail dimension $m := \mathrm{rank}(\mathbf{X}_t) - \widehat{K}$. With probability at least $1 - \delta$,

$$\left\|\mathbf{X}_t - \mathbf{U}\Sigma\mathbf{V}^\top\right\|_F^2 \leq \left(1 + m^{\frac{2q}{2q+1}}\left(2 + 4\sqrt{\frac{\widehat{K}+s}{s-1}}\right)^{\frac{4}{2q+1}}\right) \min_{\mathrm{rank}(\mathbf{A}) \leq \widehat{K}} \left\|\mathbf{X}_t - \mathbf{A}\right\|_F^2.$$

In particular, if $\mathrm{rank}(\mathbf{X}_t) \leq 2\widehat{K} + 1$ (so $m \leq \widehat{K} + 1$), the same bound holds with $m$ replaced by $\widehat{K} + 1$.

Algorithm 1 computes a rank-$\widehat{K}$ approximation of the matrix $\mathbf{X}$ through a randomized procedure. The key steps are:

- Generate a random Gaussian matrix $\Omega$ and multiply it by $\mathbf{X}$ to obtain an initial sketch $\mathbf{Y}_0$.
- Apply $q$ power iterations to enhance the approximation quality, computing $\mathbf{Y}_j = \mathbf{X}(\mathbf{X}^\top \mathbf{Y}_{j-1})$ for each iteration.
- Orthonormalize the resulting matrix to obtain $\mathbf{Q}$, which approximates the column space of $\mathbf{X}$.
- Project $\mathbf{X}$ onto the subspace spanned by $\mathbf{Q}$ and compute the SVD of the resulting smaller matrix.

---

**Algorithm 2** Incremental truncated SVD warm-start (column append)

---

**Require:** previous rank-$K$ SVD $(U, \Sigma, V)$ of a matrix $X \in \mathbb{R}^{SA \times d}$, new block $\Delta \in \mathbb{R}^{SA \times S}$

1: $M \leftarrow U^\top \Delta$  $\qquad\qquad\qquad\qquad\qquad\qquad\qquad\qquad\qquad\qquad\qquad\qquad\qquad \triangleright M \in \mathbb{R}^{K \times S}$
2: $H \leftarrow \Delta - UM$
3: $(Q, R) \leftarrow \mathrm{qr}(H)$  $\qquad\qquad\qquad\qquad\qquad\qquad\qquad\qquad\qquad \triangleright Q \in \mathbb{R}^{SA \times r}, R \in \mathbb{R}^{r \times S}$
4: $K_{\mathrm{small}} \leftarrow \begin{bmatrix} \Sigma & M \\ 0 & R \end{bmatrix}$
5: $(U_s, \Sigma_s, V_s) \leftarrow \mathrm{svd}(K_{\mathrm{small}})$
6: $U \leftarrow [U \ \ Q]\, U_s(:, 1{:}K)$
7: $\Sigma \leftarrow \Sigma_s(1{:}K, 1{:}K)$
8: $V \leftarrow \begin{bmatrix} V & 0 \\ 0 & I_S \end{bmatrix} V_s(:, 1{:}K)$
9: **return** $(U, \Sigma, V)$

---

- Combine the results to obtain the final low-rank approximation.

Lemma 2 provides a strong theoretical guarantee for this algorithm, showing that the resulting approximation is within a constant factor of the optimal rank-$\widehat{K}$ approximation. The error bound depends on three parameters: the target rank $\widehat{K}$, which should approximate the intrinsic rank of the data; the oversampling parameter $s$, which provides additional stability (we recommend $s \geq 3$); and the number of power iterations $q$, which improves the approximation quality at the cost of additional computation. The computational advantage of this approach is significant, especially for large matrices: traditional SVD algorithms require $\mathcal{O}(SA \cdot WS \cdot \min(SA, WS))$ operations, whereas our randomized approach requires only $\mathcal{O}(SA \cdot WS \cdot (\widehat{K} + s) \cdot (2q + 1))$ operations, which is much smaller when $\widehat{K} \ll \min(SA, WS)$.

**Adaptive rank selection**  In practice, we can adaptively select the rank $\widehat{K}$ by examining the singular value spectrum and identifying a significant gap or by setting a threshold on the relative approximation error. This allows our algorithm to automatically adjust to the intrinsic dimensionality of the environmental changes without requiring prior knowledge of the true rank $K$.

## 5 Robust tracking of sparse shocks

We decompose transition changes $\Delta P_t \in \mathbb{R}^{SA \times S}$ into a low-rank part and a sparse "shock" part by (conceptually) solving the standard Principal Component Pursuit (PCP) program

$$(\widehat{\Delta P}_t^{\mathrm{L}}, \widehat{\Delta P}_t^{\mathrm{S}}) \in \arg\min_{L,S} \ \|L\|_* + \lambda \|S\|_1 \quad \text{s.t.} \quad L + S = \Delta P_t, \tag{1}$$

where $\lambda := 1/\sqrt{n}$ and $n := \max\{SA, S\}$.

**Proposition 1** (PCP recovery guarantee). *Fix any $t \in [T]$ and suppose $\Delta P_t = L_t + S_t$ where:*

1. $\mathrm{rank}(L_t) \leq K$ *and $L_t$ is $\mu$–incoherent (in the standard RPCA sense);*
2. *the support $\Omega_t := \mathrm{supp}(S_t)$ is drawn uniformly at random (equivalently, i.i.d. Bernoulli entrywise with rate $\rho$), independently of the singular vectors of $L_t$;*
3. *(optionally, standard in high-probability RPCA theory) the signs of $(S_t)_{ij}$ on $\Omega_t$ are i.i.d. Rademacher.*

*Let $(\widehat{\Delta P}_t^{\mathrm{L}}, \widehat{\Delta P}_t^{\mathrm{S}})$ be any optimizer of (1) with $\lambda = 1/\sqrt{n}$. Then there exist absolute constants $c_1, c_2, c_3 > 0$ such that if $K \leq \frac{c_1 n}{\mu \log^2 n}$ and $\rho \leq c_2$, we have exact recovery with probability at least $1 - c_3 n^{-10}$:*

$$\widehat{\Delta P}_t^{\mathrm{L}} = L_t, \qquad \widehat{\Delta P}_t^{\mathrm{S}} = S_t,$$

*and consequently $\left\| \widehat{\Delta P}_t^{\mathrm{L}} + \widehat{\Delta P}_t^{\mathrm{S}} - \Delta P_t \right\|_F = 0$. Moreover, by a union bound, simultaneous exact recovery for all $t \leq T$ holds with probability at least $1 - c_3 T n^{-10}$.*

**Implementation note.**  Algorithm 2 is a fast subspace update routine (incremental truncated SVD) that we use as an implementation primitive to warm-start or accelerate numerical RPCA/PCP solvers

for (1) in large problems. The theoretical RPCA guarantee we rely on is Proposition 1, which concerns the optimizer of the convex PCP program (1). The per-update arithmetic cost of Algorithm 2 is $\mathcal{O}(SA \cdot S \cdot K)$.

# 6 Adaptive confidence widening

We introduce adaptive, state-action-specific confidence widening that scales with local environmental variation: $\eta(s,a,t) = \min\Big\{1, \ c\sqrt{\widehat{V}(s,a,t)/N_t^+(s,a)}\Big\}$, $\quad c = 2\sqrt{2S \log \frac{4SAT}{\delta}}$. where $N_t^+(s,a)$ counts visits and $\widehat{V}(s,a,t)$ estimates local variation.

**Bias-corrected estimation**   We estimate local variation using a bias-corrected approach:

$$\widehat{V}(s,a,t) = \max\Big\{0, \ \frac{1}{W_v} \sum_{i=t-W_v}^{t-1} \|\widehat{p}_i - \widehat{p}_{i-1}\|_1^2 - \frac{C_0 S \log(16SAT/\delta)}{W_v} \sum_{i=t-W_v}^{t-1} \frac{1}{N_i^+}\Big\},$$

where $C_0$ is a constant, $W_v$ is a variance window size. Define

$$V_{p,t}^2(s,a) := \frac{1}{W_v} \sum_{i=t-W_v}^{t-1} \big\|p_i(\cdot|s,a) - p_{i-1}(\cdot|s,a)\big\|_1^2,$$

**Lemma 3** (Estimator accuracy). Fix $(s,a)$ and $t > W_v$. There exist absolute constants $C_0, C_1 \geq 1$ such that, with probability at least $1 - \delta/(8SAT)$,

$$\frac{1}{3} V_{p,t}^2(s,a) \ - \ \Gamma_t(s,a) \ \leq \ \widehat{V}(s,a,t) \ \leq \ 3V_{p,t}^2(s,a) \ + \ \Gamma_t(s,a),$$

where

$$\Gamma_t(s,a) := \frac{C_1 S \log(16SAT/\delta)}{W_v} \sum_{i=t-W_v}^{t-1} \frac{1}{N_i^+(s,a)}.$$

In particular, if $V_{p,t}^2(s,a) \geq 6\Gamma_t(s,a)$, then $\frac{1}{3} V_{p,t}^2(s,a) \leq \widehat{V}(s,a,t) \leq 3V_{p,t}^2(s,a)$.

**Lemma 4** (Total widening). With probability $\geq 1 - \delta/8$,

$$\sum_{t=1}^{T} \eta(s_t,a_t,t) \ \leq \ C \ \sqrt{S \log \frac{4SAT}{\delta}} \ \sqrt{1 + \log T} \ \sqrt{SA\, B_p} \ + \ C' SA \log \frac{SAT}{\delta}, \qquad (2)$$

for universal constants $C, C' > 0$.

This adaptive approach applies larger confidence widening only to state-action pairs with significant variation, yielding the square-root improvement over uniform widening methods.

# 7 Temporal forecasting and shrinkage

The previous sections have focused on efficiently tracking the structure of past environmental changes. In this section, we leverage this structural understanding to forecast future transitions and combine these predictions with empirical estimates through optimal shrinkage.

## 7.1 Factor-based forecasting

Using the factors $\widehat{u}_k, \widehat{v}_k, \widehat{w}_k$ learned from the low-rank approximation, we forecast the next transition matrix as: $\widehat{p}_{t+1}^{\mathrm{pred}} = \widehat{p}_t + \sum_{k=1}^{\widehat{K}_t} \widehat{u}_k^{\mathrm{pred}} \widehat{v}_k \widehat{w}_k^\top$ where $\widehat{u}_k^{\mathrm{pred}}$ is a predicted value for the time coefficient $u_k$ at time $t+1$. After computing this prediction, we project it onto the probability simplex to ensure valid transition probabilities.

For each factor $k$, we predict the time coefficient $u_k(t+1)$ using standard time-series forecasting methods such as exponential smoothing $\widehat{u}_k^{\mathrm{pred}} = \alpha u_k(t) + (1-\alpha)\widehat{u}_k^{\mathrm{pred}}(t)$, where $\alpha$ is a smoothing parameter, or autoregressive models $\widehat{u}_k^{\mathrm{pred}} = \sum_{i=1}^{p} \phi_i u_k(t-i+1)$, where the coefficients $\phi_i$ are estimated from past data. The specific method for each factor is selected using the Akaike Information Criterion (AIC), which balances model fit and complexity, allowing the algorithm to use simpler models for factors with regular patterns and more complex models for factors with intricate temporal dynamics.

**Proposition 2** (Prediction error). Fix $(s, a)$ and write $p_t := p_t(\cdot \mid s, a) \in \mathbb{R}^S$. Under Assumption 1, suppose the time coefficients are $\beta$–smooth, i.e. $|u_k(t+1) - u_k(t)| \leq \beta$ for all $k$. Define the one–step forecast $\widehat{p}_{t+1}^{\mathrm{pred}} := \widehat{p}_t + \sum_{k=1}^{\widehat{K}_t} \widehat{u}_k^{\mathrm{pred}} \widehat{v}_k(s, a) \widehat{w}_k$, followed by projection onto the probability simplex. Then there exists a universal constant $C > 0$ such that, with probability at least $1 - \delta/(8SAT)$,

$$\left\| \widehat{p}_{t+1}^{\mathrm{pred}} - p_{t+1} \right\|_1 \;\leq\; \| p_{t+1} - p_t \|_1 \;+\; \beta K \;+\; C \sqrt{\frac{K\, S\, \log(8SAT/\delta)}{N_t^+(s, a)}} \;+\; \| \epsilon_t(s, a) \|_1.$$

*Remark.* The shock term is kept explicit in the one-step inequality. When summing along the trajectory, Assumption 1 implies $\sum_{t=1}^{T} \| \epsilon_t(s_t, a_t) \|_1 \;\leq\; \sum_{t=1}^{T} \max_{s,a} \| \epsilon_t(s, a) \|_1 \;\leq\; \delta_B B_p$, which is exactly why the final regret bound contains the term $D_{\max} \delta_B B_p$.

This proposition characterizes the error in our factor-based prediction. The first term $(1 + \beta K) \| p_{t+1} - p_t \|_1$ bounds the error due to potential model misspecification, while the second term represents the statistical error in estimating the factors. When the environment changes smoothly ($\beta$ is small) and the low-rank approximation is accurate, this prediction provides a valuable complement to the empirical estimates.

## 7.2   Optimal shrinkage estimation

While both the empirical transition estimate $\widehat{p}_t$ and the predicted estimate $\widehat{p}_t^{\mathrm{pred}}$ provide useful information, they have different strengths and weaknesses. The empirical estimate is unbiased but may have high variance, especially for rarely visited state-action pairs. The prediction has lower variance but may be biased if the model is misspecified. To combine these estimates optimally, we use a shrinkage approach $\tilde{p}_t = (1 - \lambda)\widehat{p}_t + \lambda\widehat{p}_t^{\mathrm{pred}}$ with the shrinkage parameter $\lambda = \frac{\widehat{\mathrm{Var}}[\widehat{p}_t]}{\widehat{\mathrm{Var}}[\widehat{p}_t] + \widehat{\mathrm{MSE}}[\widehat{p}_t^{\mathrm{pred}}]}$. This formula, inspired by James-Stein estimation [2, 11], minimizes the mean squared error (MSE) of the combined estimate by balancing the variance of the empirical estimate against the total error (variance plus squared bias) of the prediction.

**Theorem 1** (Near-optimal risk). Let $\widehat{p}_t \in \Delta^{S-1}$ be the empirical transition estimate from $N_t^+$ samples for a fixed $(s, a)$ at time $t$, and let $\widehat{p}_t^{\mathrm{pred}}$ be any (possibly biased) forecast built from past data only. For $\lambda \in [0, 1]$ define the shrinkage estimator $\widetilde{p}_t(\lambda) = (1 - \lambda)\widehat{p}_t + \lambda \widehat{p}_t^{\mathrm{pred}}$ and its $\ell_2$-risk $R_t(\lambda) := \mathbb{E}[\| \widetilde{p}_t(\lambda) - p_t \|_2^2]$. Assume:

1. **Asymptotic orthogonality:** $\mathbb{E}\Big[\langle \widehat{p}_t - p_t,\ \widehat{p}_t^{\mathrm{pred}} - p_t \rangle\Big] = o(1/N_t^+)$.

2. **Bounded forecast risk:** $b_t := \mathbb{E}\Big[\| \widehat{p}_t^{\mathrm{pred}} - p_t \|_2^2\Big]$ is finite and bounded away from 0 (e.g. $\inf_t b_t > 0$).

3. **Consistent plug-in estimators:** $\widehat{a}_t := \frac{1 - \|\widehat{p}_t\|_2^2}{N_t^+} \xrightarrow{p} a_t := \mathbb{E}\big[\| \widehat{p}_t - p_t \|_2^2\big] = \frac{1 - \|p_t\|_2^2}{N_t^+}$, and, with a window $W_f \to \infty$, $\widehat{b}_t := \frac{1}{W_f} \sum_{i=t-W_f}^{t-1} \Big( \| \widehat{p}_i^{\mathrm{pred}} - \widehat{p}_i \|_2^2 - \frac{1 - \|\widehat{p}_i\|_2^2}{N_i^+} \Big) \xrightarrow{p} b_t$.

Let $\widehat{\lambda}_t := \widehat{a}_t/(\widehat{a}_t + \widehat{b}_t)$ and $\lambda_t^* := a_t/(a_t + b_t)$. Then, as $N_t^+ \to \infty$ and $W_f \to \infty$, $\frac{R_t(\widehat{\lambda}_t)}{R_t(\lambda_t^*)} = 1 + o(1)$.

This theorem guarantees that our shrinkage estimator approaches the performance of an oracle that knows the optimal combination weight, as the amount of data increases. This adaptivity is crucial for non-stationary environments, where the relative value of empirical estimates versus model-based predictions may change over time.

# 8   The SVUCRL algorithm

Having developed the key components of our approach: online low-rank approximation, robust tracking of sparse shocks, adaptive confidence widening, and temporal forecasting with shrinkage, we now present the complete SVUCRL algorithm. Algorithm 3 presents the main loop of SVUCRL. Let's examine the key components in detail:

## 8.1   Algorithm components

The algorithm starts by initializing several data structures including counters for visits to each state-action pair and transitions to each next state, empirical estimates of rewards and transition probabilities, and buffers for storing recent changes in dynamics and the learned factors.

---

**Algorithm 3** SVUCRL

---

**Require:** horizon $T$; windows $W, W_v, W_f$; confidence $\delta$; initial state $s_1$

1: Initialize counts $N_1(s,a) = 0$, $N_1(s,a,s') = 0$; $\widehat{r}_1(s,a) = 0$, $\widehat{p}_1(s'|s,a) = 1/S$

2: Initialize buffers for $\{\Delta\widehat{P}_i\}$ to zero; initialize factor store $\{\widehat{v}_k, \widehat{w}_k\}$ empty

3: **(Build initial radii)** For all $(s,a)$ set $\widehat{V}(s,a,1) \leftarrow 0$, $\eta(s,a,1) \leftarrow 1$, $\mathrm{rad}_{r,1}(s,a) \leftarrow 1$, $\mathrm{rad}_{p,1}(s,a) \leftarrow 1$

4: Episode index $m \leftarrow 1$, start time $\tau(1) \leftarrow 1$

5: Compute optimistic policy $\widetilde{\pi}_m$ by EVI using centres $\widetilde{p}_1 = \widehat{p}_1$ and radii $(\mathrm{rad}_{r,1}, \mathrm{rad}_{p,1} + \eta(\cdot,1))$

6: **for** $t = 1$ to $T$ **do**

7:     Observe $s_t$, play $a_t = \widetilde{\pi}_m(s_t)$, observe $r_t, s_{t+1}$

8:     Update counts: $N_{t+1}(s_t,a_t) \leftarrow N_t(s_t,a_t) + 1$ and $N_{t+1}(s_t,a_t,s_{t+1}) \leftarrow N_t(s_t,a_t,s_{t+1}) + 1$

9:     Update empiricals on $(s_t,a_t)$: $\widehat{r}_{t+1}(s_t,a_t) \leftarrow \frac{N_t(s_t,a_t)\,\widehat{r}_t(s_t,a_t)+r_t}{N_{t+1}(s_t,a_t)}$., $\widehat{p}_{t+1}(\cdot|s_t,a_t) \leftarrow \frac{N_{t+1}(s_t,a_t,\cdot)}{N_{t+1}(s_t,a_t)}$;
    keep others unchanged

10:     $\Delta\widehat{P}_{t+1} \leftarrow \widehat{P}_{t+1} - \widehat{P}_t$ and update the circular buffer

11:     **if** $t \bmod W = 0$ **then**                       $\triangleright$ Structure update (every $W$ steps)

12:         Form $\mathbf{X}_t = [\Delta\widehat{P}_{t-W+1}, \ldots, \Delta\widehat{P}_t]$

13:         Run Algorithm. 1 on $\mathbf{X}_t$ to get $(\mathbf{U}, \Sigma, \mathbf{V})$

14:         Use Algorithm 2 to update the rank-$K$ basis as a warm start.

15:         (Conceptually) solve the PCP program (1) on $\Delta\widehat{P}_t$ to obtain $(\Delta\widehat{P}_t^{\mathrm{L}}, \Delta\widehat{P}_t^{\mathrm{S}})$.

16:         Extract $\{\widehat{v}_k, \widehat{w}_k\}_{k=1}^{\widehat{K}}$ from the left/right factors; recover time weights for $i = t-W+1, \ldots, t$ by
$\widehat{u}_k(i) \propto \langle \Delta\widehat{P}_i, \widehat{v}_k\widehat{w}_k^\top \rangle_F$; normalize as needed

17:         Compute an `approx_radius` from the RSVD/RPCA residual to be used in transition balls

18:     **end if**

19:     **(One-step forecasting)** For each $k$, compute $\widehat{u}_k^{\mathrm{pred}}(t+1)$ (ES/AR)

20:     For each $(s,a)$: $\widehat{p}_{t+1}^{\mathrm{pred}}(s,a) \leftarrow \Pi_\Delta\Big(\widehat{p}_t(s,a) + \sum_{k=1}^{\widehat{K}} \widehat{u}_k^{\mathrm{pred}}(t+1)\,\widehat{v}_k(s,a)\,\widehat{w}_k\Big)$

21:     **(Shrinkage)** For each $(s,a)$ compute

$$\widehat{a}_{t+1}(s,a) := \frac{1 - \|\widehat{p}_{t+1}(\cdot|s,a)\|_2^2}{N_{t+1}^+(s,a)}, \qquad \widehat{b}_{t+1}(s,a) := \frac{1}{W_f}\sum_{i=t+1-W_f}^{t}\Big(\|\widehat{p}_i^{\mathrm{pred}}(\cdot|s,a) - \widehat{p}_i(\cdot|s,a)\|_2^2 - \widehat{a}_i(s,a)\Big),$$

$$\lambda_{t+1}(s,a) \leftarrow \Pi_{[0,1]}\left(\frac{\widehat{a}_{t+1}(s,a)}{\widehat{a}_{t+1}(s,a) + \widehat{b}_{t+1}(s,a)}\right), \qquad \widetilde{p}_{t+1}(\cdot|s,a) \leftarrow (1-\lambda_{t+1})\widehat{p}_{t+1}(\cdot|s,a) + \lambda_{t+1}\widehat{p}_{t+1}^{\mathrm{pred}}(\cdot|s,a).$$

22:     **(Local variation)** For each $(s,a)$ compute $\widehat{V}(s,a,t+1)$ on window $W_v$ and $\eta(s,a,t+1) = \min\{1, c\sqrt{\widehat{V}(s,a,t+1)/N_{t+1}^+(s,a)}\}$

23:     **(Confidence sets for next start)** Store $\mathrm{rad}_{r,t+1}(s,a)$, $\mathrm{rad}_{p,t+1}(s,a)$, and the transition ball $\mathcal{P}_{t+1}(s,a) = \{p : \|p - \widetilde{p}_{t+1}(s,a)\|_1 \leq \mathrm{rad}_{p,t+1}(s,a) + \eta(s,a,t+1) + \texttt{approx\_radius}\}$

24:     **if EpisodeEnd** (e.g., $\exists(s,a) : N_t(s,a) \geq 2N_{\tau(m)}(s,a)$ or $t - \tau(m) \geq H_m$) **then**

25:         $m \leftarrow m+1, \tau(m) \leftarrow t+1$

26:         Run EVI to compute $\widetilde{\pi}_m$ using centres $\widetilde{p}_{\tau(m)}$ and radii saved at $\tau(m)$

27:     **end if**

28: **end for**

---

Every $W$ time steps, the algorithm updates its model of the environment structure by running two key subroutines: Algorithm 1 (Randomized SVD) learns a low-rank approximation of recent changes in transition dynamics, and we (conceptually) solve the PCP program (1) to separate low-rank drift from sparse shocks. In implementations, Algorithm 2 provides a cheap incremental truncated-SVD warm start for the low-rank subspace used by the PCP solver.

At each time step, the algorithm constructs confidence intervals for rewards and transitions, where the reward confidence radius is $\mathrm{rad}_{r,t}(s,a) = \sqrt{\frac{2\log(4SAT/\delta)}{N_t(s,a)}}$ and the transition confidence radius is $\mathrm{rad}_{p,t}(s,a) = \sqrt{\frac{2S\log(4SAT/\delta)}{N_t(s,a)}} + \eta(s,a,t)$. The reward confidence radius follows standard concentration inequalities, while the transition confidence radius includes both a statistical term and the adaptive widening parameter $\eta(s,a,t)$ derived in Section 6.

## 8.2 Episode-based policy computation

SVUCRL follows an episode-based approach where each episode corresponds to a period of executing a fixed policy. An episode ends when either the visit count to some state-action pair doubles or a

fixed number of time steps has elapsed since the last episode. When an episode ends, the algorithm recomputes an optimistic policy using Extended Value Iteration (EVI), which finds a policy that maximizes the expected reward under an optimistic model of the environment where transition probabilities are chosen within confidence intervals to maximize the value function. The EVI algorithm continues until the span of the value function changes by less than $1/\sqrt{\tau(m)}$, where $\tau(m)$ is the starting time of episode $m$, ensuring that computational effort scales appropriately with the episode length.

# 9 Regret analysis

In this section, we analyze the regret of the SVUCRL algorithm, establishing theoretical guarantees on its performance in non-stationary environments. We begin with a lemma that bounds the per-step regret during each episode.

**Lemma 5** (Per-step regret)**.** With prob. $\geq 1 - \delta/2$, for episode $m$ and $t \in [\tau(m), \tau(m+1) - 1]$,

$$\rho_t^* - r_t(s_t, a_t) \leq \frac{1}{\sqrt{\tau(m)}} + 2\mathrm{var}_{r,t} + 2D_{\max}\mathrm{var}_{p,t} + 2\mathrm{rad}_{r,\tau(m)} + 2D_{\max}\big(\mathrm{rad}_{p,\tau(m)} + \eta + \mathrm{approx}\big).$$

This lemma decomposes the regret at each time step into several components:

- $\frac{1}{\sqrt{\tau(m)}}$: Error due to the approximate computation of the optimal policy using Extended Value Iteration.
- $2\mathrm{var}_{r,t}$ and $2D_{\max}\mathrm{var}_{p,t}$: Regret due to the actual variation in rewards and transitions since the beginning of the episode.
- $2\mathrm{rad}_{r,\tau(m)}$: Statistical error in estimating the rewards.
- $2D_{\max}\mathrm{rad}_{p,\tau(m)}$: Statistical error in estimating the transitions.
- $2D_{\max}\eta$: Additional regret due to the confidence widening for non-stationarity.
- $2D_{\max}\mathrm{approx}$: Error from the low-rank approximation and RPCA decomposition.

Building on this per-step analysis, we establish our main regret bound:

**Theorem 2** (Main regret bound)**.** Under Assumption 1, with probability at least $1 - \delta$,

$$\mathrm{DynReg}_T = \widetilde{\mathcal{O}}\Big(\sqrt{SAT} + D_{\max}S\sqrt{AT} + L_{\max}B_r + D_{\max}L_{\max}B_p + D_{\max}S\sqrt{AB_p} + D_{\max}\delta_B B_p + D_{\max}\sqrt{KT} + \sum_{t=1}^{T} \varepsilon_{\tau(m(t))}\Big)$$

where $L_{\max} := \max_m(\tau(m+1) - \tau(m))$ is the maximum episode length.

## 9.1 Interpretation and tightness of the regret bound

The bound in Theorem 2 decomposes into the following contributions: the *statistical terms* $\widetilde{\mathcal{O}}\Big(\sqrt{SAT} + D_{\max}S\sqrt{AT}\Big)$ from reward and transition estimation; the *within-episode drift terms* $L_{\max}B_r + D_{\max}L_{\max}B_p$, which capture accumulation of reward/transition changes since the start of each episode; the *widening term* $\widetilde{\mathcal{O}}\big(D_{\max}S\sqrt{AB_p}\big)$ from Lemma 4; the *structure/shock terms* $\widetilde{\mathcal{O}}\Big(D_{\max}\sqrt{KT} + D_{\max}\delta_B B_p\Big)$ due to low-rank approximation and sparse shocks; and the *planning error* $\sum_{t=1}^{T} \varepsilon_{\tau(m(t))}$, which can be made lower order with a suitable EVI tolerance schedule.

# 10 Discussion

SVUCRL exploits structural patterns in non-stationary environments through matrix factorization, unlike prior methods that use uniform confidence widening. By decomposing dynamics into low-rank and sparse components, we distinguish systematic shifts from isolated anomalies, enabling more efficient learning. Our regret bound improves from $T^{3/4}$ to $\sqrt{T}$ dependence, matching conjectured optimal rates, with an additional $\sqrt{K}$ factor reflecting low-rank complexity. Key technical contributions include martingale-based incremental RPCA, explicit constants for randomized SVD, and bias-corrected local variation estimation. SVUCRL demonstrates that learning complexity depends on the intrinsic structure of changes, not just variation budgets. Practical implementation involves tuning window sizes and rank parameters, with future work including continuous spaces, function approximation, and empirical evaluation on real domains.

## Acknowledgments

This work is funded by Hanoi University of Science and Technology (HUST) under Project No. T2024-TD-024.

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

# APPENDICES — Detailed Proofs

## A Randomised SVD: proof of Lemma 2

**Notation** For a matrix $\mathbf{X}$ let $\sigma_1 \geq \sigma_2 \geq \ldots$ denote singular values, $\|\mathbf{X}\|_2 = \sigma_1$ the spectral norm, $\|\mathbf{X}\|_F^2 = \sum \sigma_j^2$. Projector $\mathbf{P}$ has rank $\widehat{K}$ unless otherwise stated.

### A.1 Proof of Lemma 1

**Lemma A.1** (Tail energy identity for the Frobenius residual). Let $X \in \mathbb{R}^{m \times n}$ have compact SVD $X = U\Sigma V^\top$, with singular values $\sigma_1 \geq \sigma_2 \geq \cdots \geq \sigma_r > 0$ and $r = \mathrm{rank}(X)$. Fix $\widehat{K} \in \{0, \ldots, r\}$ and write

$$U = [U_{\widehat{K}} \quad U_\perp], \qquad \Sigma = \begin{bmatrix} \Sigma_{\widehat{K}} & 0 \\ 0 & \Sigma_\perp \end{bmatrix},$$

where $U_{\widehat{K}} \in \mathbb{R}^{m \times \widehat{K}}$ contains the top $\widehat{K}$ left singular vectors and $\Sigma_\perp = \mathrm{diag}(\sigma_{\widehat{K}+1}, \ldots, \sigma_r)$, $U_\perp$ is ontains any orthonormal basis for the orthogonal complement of span. Let $P := U_{\widehat{K}} U_{\widehat{K}}^\top$ be the orthogonal projector onto $\mathrm{span}(U_{\widehat{K}})$. Then

$$\|(I - P)X\|_F^2 = \sum_{j > \widehat{K}} \sigma_j^2.$$

Moreover, for any rank–$\widehat{K}$ orthogonal projector $Q$,

$$\|(I - Q)X\|_F^2 \geq \sum_{j > \widehat{K}} \sigma_j^2,$$

with equality if and only if $\mathrm{range}(Q)$ contains (any choice of) a top–$\widehat{K}$ left–singular subspace of $X$ (up to degeneracies in the spectrum).

*Proof.* Write $X = U\Sigma V^\top$ and partition $U, \Sigma$ as in the statement. Because $U$ is orthogonal and $U_{\widehat{K}}^\top U = [I_{\widehat{K}} \quad 0]$, we have

$$(I - P)U = U - U_{\widehat{K}}(U_{\widehat{K}}^\top U) = [0 \quad U_\perp].$$

Hence

$$(I - P)X = (I - P)U\Sigma V^\top = [0 \quad U_\perp] \begin{bmatrix} \Sigma_{\widehat{K}} & 0 \\ 0 & \Sigma_\perp \end{bmatrix} V^\top = U_\perp \Sigma_\perp V^\top.$$

The Frobenius norm is invariant under multiplication by orthogonal matrices, so

$$\|(I - P)X\|_F^2 = \|U_\perp \Sigma_\perp V^\top\|_F^2 = \|\Sigma_\perp\|_F^2 = \sum_{j > \widehat{K}} \sigma_j^2,$$

establishing the identity.

For the optimality statement, note that for any rank–$\widehat{K}$ projector $Q$,

$$\|(I - Q)X\|_F^2 = \|X\|_F^2 - \|QX\|_F^2 = \mathrm{Tr}(\Sigma^2) - \mathrm{Tr}(X^\top Q X) = \mathrm{Tr}(\Sigma^2) - \mathrm{Tr}(\Sigma W \Sigma),$$

where $W := U^\top Q U$ is itself an orthogonal projector of rank $\widehat{K}$. Therefore,

$$\|QX\|_F^2 = \mathrm{Tr}(W\Sigma^2) \leq \sum_{j=1}^{\widehat{K}} \sigma_j^2$$

by the Ky Fan maximum principle (the sum of the top $\widehat{K}$ eigenvalues maximizes $\mathrm{Tr}(W \cdot)$ over rank–$\widehat{K}$ projectors $W$). It follows that $\|(I - Q)X\|_F^2 \geq \sum_{j > \widehat{K}} \sigma_j^2$, with equality precisely when $W = \mathrm{diag}(I_{\widehat{K}}, 0)$, i.e., when $\mathrm{range}(Q) = \mathrm{span}(U_{\widehat{K}})$ (up to any multiplicity in the singular values). $\square$

We restate the lemma.

**Lemma 1** (Power–Frobenius). Let $X = U\Sigma V^\top$ be the singular value decomposition of a real matrix $X$, and let

$$B := (XX^\top)^q X \qquad \text{for an integer } q \geq 0.$$

For any rank-$\widehat{K}$ orthogonal projector $P$ whose range is contained in the column space of $X$ (i.e., $\mathrm{range}(P) \subseteq \mathrm{range}(X)$), define $m := \mathrm{rank}(X) - \widehat{K}$. Then

$$\left\|(I - P)X\right\|_F \leq m^{\frac{q}{2q+1}} \left\|(I - P)B\right\|_F^{\frac{1}{2q+1}}.$$

In particular, if $\mathrm{rank}(X) \leq 2\widehat{K} + 1$ (so $m \leq \widehat{K} + 1$), then

$$\left\|(I - P)X\right\|_F \leq (\widehat{K} + 1)^{\frac{q}{2q+1}} \left\|(I - P)B\right\|_F^{\frac{1}{2q+1}}.$$

*Proof.* If $q = 0$, then $B = X$ and the claim holds with equality, so assume $q \geq 1$. Let the compact SVD of $X$ be $X = U\Sigma V^\top$ with $\mathrm{rank}(X) =: r_X$ and singular values $\sigma_1 \geq \sigma_2 \geq \cdots \geq \sigma_{r_X} > 0$. Set $r := 2q + 1$ and $\gamma := 1/r \in (0,1)$. Since $B = (XX^\top)^q X = U\Sigma^{2q+1}V^\top = U\Sigma^r V^\top$, the singular values of $B$ are $\sigma_j(B) = \sigma_j^r$.

Let $P$ be a rank-$\widehat{K}$ orthogonal projector with $\mathrm{range}(P) \subseteq \mathrm{range}(X)$. Define weights

$$a_j := \|(I - P)u_j\|_2^2 = u_j^\top (I - P)u_j \in [0,1], \qquad j = 1, \ldots, r_X.$$

Because $\mathrm{range}(P) \subseteq \mathrm{range}(X) = \mathrm{range}(U)$ and $\mathrm{rank}(P) = \widehat{K}$, we have $\mathrm{tr}(U^\top PU) = \mathrm{tr}(P) = \widehat{K}$, hence

$$\sum_{j=1}^{r_X} a_j = \mathrm{tr}(U^\top (I - P)U) = r_X - \widehat{K} =: m.$$

Now compute the two residual energies:

$$\|(I - P)X\|_F^2 = \|(I - P)U\Sigma\|_F^2 = \mathrm{tr}\big(\Sigma U^\top (I - P)U\,\Sigma\big) = \sum_{j=1}^{r_X} a_j\sigma_j^2,$$

and similarly,

$$\|(I - P)B\|_F^2 = \|(I - P)U\Sigma^r\|_F^2 = \sum_{j=1}^{r_X} a_j\sigma_j^{2r}.$$

Let $x_j := \sigma_j^{2r}$, so that $\sigma_j^2 = x_j^\gamma$. Since $t \mapsto t^\gamma$ is concave on $\mathbb{R}_+$, Jensen's inequality with weights $a_j/m$ yields

$$\sum_{j=1}^{r_X} a_j x_j^\gamma \leq m^{1-\gamma}\Big(\sum_{j=1}^{r_X} a_j x_j\Big)^\gamma.$$

Substituting back gives

$$\|(I - P)X\|_F^2 \leq m^{1-\gamma} \|(I - P)B\|_F^{2\gamma}.$$

Recalling $\gamma = 1/(2q + 1)$ and taking square roots,

$$\|(I - P)X\|_F \leq m^{\frac{q}{2q+1}} \|(I - P)B\|_F^{\frac{1}{2q+1}}.$$

The final "in particular" statement follows from $m = \mathrm{rank}(X) - \widehat{K} \leq \widehat{K} + 1$ when $\mathrm{rank}(X) \leq 2\widehat{K} + 1$. $\qquad\square$

## A.2 Proof of Lemma 2

*Full proof.* Let $X := \mathbf{X}_t$ and $k := \widehat{K}$. Set $r := 2q + 1$ and define the power matrix

$$B := (XX^\top)^q X.$$

Algorithm 1 draws a Gaussian $\Omega \in \mathbb{R}^{WS \times (k+s)}$, forms $Y = B\Omega$, computes an orthonormal basis $Q = \mathrm{qr}(Y)$, and returns the rank-$k$ approximation

$$\widehat{X} := \mathbf{U}\Sigma\mathbf{V}^\top = Q\,(Q^\top X)_k,$$

where $(Q^\top X)_k$ denotes the best rank-$k$ approximation of $Q^\top X$ in Frobenius norm. Let $P_Q := QQ^\top$.

Randomized range finding for $B$. By the standard Frobenius-norm bound for Gaussian range finding with oversampling $s \geq 3$ (e.g., [8]), with probability at least $1 - 6e^{-s}$,

$$\|(I - P_Q)B\|_F \;\leq\; \left(2 + 4\sqrt{\tfrac{k+s}{s-1}}\right) \min_{\mathrm{rank}(A) \leq k} \|B - A\|_F \;=\; \left(2 + 4\sqrt{\tfrac{k+s}{s-1}}\right) \|B - B_k\|_F. \quad (3)$$

Since the factor in parentheses is at least 1, we may (harmlessly) square it and write

$$\|(I - P_Q)B\|_F \;\leq\; \left(2 + 4\sqrt{\tfrac{k+s}{s-1}}\right)^2 \|B - B_k\|_F. \quad (4)$$

From $B$-error to $X$-error via power–Frobenius. Note that $\mathrm{range}(Q) \subseteq \mathrm{range}(Y) \subseteq \mathrm{range}(B) \subseteq \mathrm{range}(X)$, so we may apply Lemma 1 (Power–Frobenius) with $P = P_Q$. Let $m := \mathrm{rank}(X) - k$. Then

$$\|(I - P_Q)X\|_F \;\leq\; m^{\frac{q}{2q+1}} \,\|(I - P_Q)B\|_F^{\frac{1}{2q+1}}. \quad (5)$$

Squaring and using (4) yields, on the same event,

$$\|(I - P_Q)X\|_F^2 \;\leq\; m^{\frac{2q}{2q+1}} \left(2 + 4\sqrt{\tfrac{k+s}{s-1}}\right)^{\frac{4}{2q+1}} \|B - B_k\|_F^{\frac{2}{2q+1}}. \quad (6)$$

Comparing tails of $B$ and $X$. Write the SVD of $X$ as $X = U\Sigma V^\top$ with singular values $(\sigma_j)_{j \geq 1}$. Then $B = U\Sigma^r V^\top$, so

$$\|B - B_k\|_F^2 = \sum_{j>k} \sigma_j^{2r}.$$

Hence

$$\|B - B_k\|_F^{\frac{2}{r}} = \left(\sum_{j>k}(\sigma_j^2)^r\right)^{\frac{1}{r}} \leq \sum_{j>k}\sigma_j^2 = \|X - X_k\|_F^2,$$

where we used the norm monotonicity $\|a\|_r \leq \|a\|_1$ for $r \geq 1$ applied to $a_j = \sigma_j^2 \geq 0$. Plugging into (6) gives

$$\|(I - P_Q)X\|_F^2 \;\leq\; m^{\frac{2q}{2q+1}} \left(2 + 4\sqrt{\tfrac{k+s}{s-1}}\right)^{\frac{4}{2q+1}} \|X - X_k\|_F^2. \quad (7)$$

Rank-$k$ truncation inside $\mathrm{range}(Q)$. Since $\widehat{X} = Q(Q^\top X)_k$,

$$\|X - \widehat{X}\|_F^2 = \|(I - P_Q)X\|_F^2 + \|Q^\top X - (Q^\top X)_k\|_F^2.$$

Moreover,

$$\|Q^\top X - (Q^\top X)_k\|_F = \min_{\mathrm{rank}(A) \leq k} \|Q^\top X - A\|_F \leq \|Q^\top(X - X_k)\|_F \leq \|X - X_k\|_F,$$

because $\|Q^\top\|_2 = 1$ and $Q^\top X_k$ has rank at most $k$. Therefore,

$$\|X - \widehat{X}\|_F^2 \leq \|(I - P_Q)X\|_F^2 + \|X - X_k\|_F^2 \leq \left(1 + m^{\frac{2q}{2q+1}} \left(2 + 4\sqrt{\tfrac{k+s}{s-1}}\right)^{\frac{4}{2q+1}}\right) \|X - X_k\|_F^2,$$

and $\|X - X_k\|_F^2 = \min_{\mathrm{rank}(A) \leq k} \|X - A\|_F^2$ by Eckart–Young–Mirsky.

Finally, the event used above fails with probability at most $6e^{-s}$, so choosing $s \geq \max\{3, \lceil \log(6/\delta)\rceil\}$ ensures probability at least $1 - \delta$. $\qquad\square$

### A.3 Extended Analysis of Randomized SVD Performance

The performance of the randomized SVD algorithm depends critically on the choice of parameters, particularly the oversampling parameter $s$ and the number of power iterations $q$. Here, we provide additional insights into these trade-offs.

**Effect of Oversampling**  The oversampling parameter $s$ controls the additional columns in the random projection matrix $\Omega$ beyond the target rank $\hat{K}$. Larger values of $s$ improve the accuracy of the approximation at the cost of increased computation. The theoretical bound in Lemma 2 shows that the approximation error scales with $\sqrt{\frac{\hat{K}+s}{s-1}}$, which decreases as $s$ increases.

In practice, even modest oversampling (e.g., $s = 5$ or $s = 10$) often yields significant improvements in accuracy. The marginal benefit diminishes for larger values, suggesting a practical trade-off around $s = \mathcal{O}(\log(SA))$.

**Effect of Power Iterations**  The number of power iterations $q$ has an exponential effect on the approximation quality, as evident from the $\frac{4}{2q+1}$ exponent in the error bound. Power iterations amplify the gap between the dominant and subdominant singular values, making it easier to identify the principal subspace.

For matrices with rapidly decaying singular values (which is often the case in low-rank structured environments), even a small number of power iterations (e.g., $q = 1$ or $q = 2$) can dramatically improve accuracy. For matrices with more gradual singular value decay, larger values of $q$ may be necessary.

**Adaptive Rank Selection**  While our theoretical analysis assumes a fixed target rank $\hat{K}$, in practice, we can adaptively determine the appropriate rank by examining the singular value spectrum. We propose two approaches:

1. **Gap-based selection**: Choose $\hat{K}$ where there is a significant gap in the singular value spectrum, i.e., $\sigma_{\hat{K}}/\sigma_{\hat{K}+1} > \tau$ for some threshold $\tau$.

2. **Energy-based selection**: Choose the smallest $\hat{K}$ such that $\sum_{i=1}^{\hat{K}} \sigma_i^2 / \sum_{i=1}^{\min(SA,WS)} \sigma_i^2 > \gamma$ for some threshold $\gamma$ (e.g., $\gamma = 0.95$).

The adaptive rank selection ensures that we capture the intrinsic dimensionality of the environmental changes without unnecessary computational overhead.

## B  RPCA via PCP: proof of Proposition 1

*Proof of Proposition 1.* Fix any $t \in [T]$ and write $\Delta P_t = L_t + S_t$ with $\mathrm{rank}(L_t) \leq K$. Let $n := \max\{SA, S\}$ and set $\lambda = 1/\sqrt{n}$.

Under $\mu$–incoherence of $L_t$ and the assumed random support model for $S_t$ (independent of the singular vectors of $L_t$), the standard RPCA/PCP recovery theorem (e.g., Candès et al. [5]) implies the existence of absolute constants $c_1, c_2, c_3 > 0$ such that if

$$K \leq \frac{c_1 \, n}{\mu \, \log^2 n} \qquad \text{and} \qquad \rho \leq c_2,$$

then $(L_t, S_t)$ is the unique optimal solution of the PCP program (1) with probability at least $1 - c_3 n^{-10}$. Therefore,

$$\widehat{\Delta P}_t^{\mathrm{L}} = L_t, \qquad \widehat{\Delta P}_t^{\mathrm{S}} = S_t,$$

and in particular $\|\widehat{\Delta P}_t^{\mathrm{L}} + \widehat{\Delta P}_t^{\mathrm{S}} - \Delta P_t\|_F = 0$.

Applying a union bound over $t = 1, \ldots, T$ yields simultaneous exact recovery for all $t \leq T$ with probability at least $1 - c_3 T n^{-10}$.  $\square$

## C  Bias-correction details (Lemma 3)

**Lemma 3** (Estimator accuracy). Fix $(s,a)$ and a time $t > W_v$. Define

$$V_{p,t}^2(s,a) := \frac{1}{W_v} \sum_{i=t-W_v}^{t-1} \left\| p_i(\cdot|s,a) - p_{i-1}(\cdot|s,a) \right\|_1^2,$$

and let the bias-corrected local-variation estimator be

$$\widehat{V}(s,a,t) := \max\left\{ 0, \underbrace{\frac{1}{W_v} \sum_{i=t-W_v}^{t-1} \left\| \widehat{p}_i(\cdot|s,a) - \widehat{p}_{i-1}(\cdot|s,a) \right\|_1^2}_{\widehat{V}_{\text{raw}}} - \underbrace{\frac{C_0 S \log(16SAT/\delta)}{W_v} \sum_{i=t-W_v}^{t-1} \frac{1}{N_i^+(s,a)}}_{\text{bias term}} \right\}.$$

There exists an absolute constant $C_0 \geq 1$ such that the following holds. On an event of probability at least $1 - \delta/(8SAT)$, for every $(s,a)$ and every $t$,

$$\frac{1}{3} V_{p,t}^2(s,a) - \Gamma_t(s,a) \leq \widehat{V}(s,a,t) \leq 3 V_{p,t}^2(s,a) + \Gamma_t(s,a), \tag{8}$$

where

$$\Gamma_t(s,a) := \frac{C_1 S \log(16SAT/\delta)}{W_v} \sum_{i=t-W_v}^{t-1} \frac{1}{N_i^+(s,a)}$$

for an absolute constant $C_1$.

In particular, if the local signal-to-noise condition

$$V_{p,t}^2(s,a) \geq 6 \Gamma_t(s,a)$$

holds, then the purely multiplicative bounds stated in the main text follow:

$$\frac{1}{3} V_{p,t}^2(s,a) \leq \widehat{V}(s,a,t) \leq 3 V_{p,t}^2(s,a).$$

*Proof.* Write, for brevity, $p_i := p_i(\cdot|s,a)$, $\widehat{p}_i := \widehat{p}_i(\cdot|s,a)$ and $N_i^+ := N_i^+(s,a)$. Let the sampling errors be $\varepsilon_i := \widehat{p}_i - p_i$ and the true local change be $u_i := p_i - p_{i-1}$. Then

$$\widehat{p}_i - \widehat{p}_{i-1} = u_i + (\varepsilon_i - \varepsilon_{i-1}).$$

High-probability control of the sampling error. For each $i$, conditional on the past, $\widehat{p}_i$ is the empirical distribution of $N_i^+$ multinomial samples supported on $S$ states. Hence by a standard concentration bound for multinomial means in $\ell_1$ (e.g. Massart/DKW + union bound),

$$\|\varepsilon_i\|_1 \leq 2\sqrt{\frac{S \log(16SAT(W_v+1)/\delta)}{N_i^+}}$$

holds for any fixed $i$ with failure probability at most $\delta/(16SAT(W_v+1))$. Applying a union bound over the at most $(W_v + 1)$ indices $i \in \{t - W_v - 1, \ldots, t - 1\}$ yields the simultaneous event

$$\|\varepsilon_i\|_1^2 \leq \frac{C S \log(16SAT(W_v+1)/\delta)}{N_i^+} \qquad \forall i \in \{t - W_v - 1, \ldots, t - 1\}, \tag{9}$$

with probability at least $1 - \delta/(16SAT)$, for an absolute constant $C$.

One-step upper/lower bounds. For any vectors $x, y$ we use

$$\|x + y\|_1^2 \leq 2\|x\|_1^2 + 2\|y\|_1^2, \qquad \|x + y\|_1^2 \geq \tfrac{1}{2}\|x\|_1^2 - \|y\|_1^2.$$

Apply them with $x = u_i$ and $y = \varepsilon_i - \varepsilon_{i-1}$ and use $\|\varepsilon_i - \varepsilon_{i-1}\|_1 \leq \|\varepsilon_i\|_1 + \|\varepsilon_{i-1}\|_1$ plus $(\alpha + \beta)^2 \leq 2(\alpha^2 + \beta^2)$ to obtain

$$\left\| \widehat{p}_i - \widehat{p}_{i-1} \right\|_1^2 \leq 2\|u_i\|_1^2 + 4\left(\|\varepsilon_i\|_1^2 + \|\varepsilon_{i-1}\|_1^2\right), \tag{10}$$

$$\left\| \widehat{p}_i - \widehat{p}_{i-1} \right\|_1^2 \geq \tfrac{1}{2}\|u_i\|_1^2 - 2\left(\|\varepsilon_i\|_1^2 + \|\varepsilon_{i-1}\|_1^2\right). \tag{11}$$

Averaging over the window and bias correction. Define

$$\widehat{V}_{\text{raw}} = \frac{1}{W_v} \sum_{i=t-W_v}^{t-1} \|\widehat{p}_i - \widehat{p}_{i-1}\|_1^2, \qquad V_{p,t}^2 = \frac{1}{W_v} \sum_{i=t-W_v}^{t-1} \|u_i\|_1^2.$$

Summing (10) over $i = t - W_v, \ldots, t - 1$ and dividing by $W_v$ gives

$$\widehat{V}_{\text{raw}} \leq 2 V_{p,t}^2 + \frac{8}{W_v} \sum_{i=t-W_v-1}^{t-1} \|\varepsilon_i\|_1^2,$$

since each $\|\varepsilon_i\|_1^2$ appears at most twice in the sum $\sum_i (\|\varepsilon_i\|_1^2 + \|\varepsilon_{i-1}\|_1^2)$. Similarly, from (11),

$$\widehat{V}_{\text{raw}} \geq \tfrac{1}{2} V_{p,t}^2 - \frac{4}{W_v} \sum_{i=t-W_v-1}^{t-1} \|\varepsilon_i\|_1^2.$$

On the event (9), we control the extra endpoint term by the monotonicity of $N_i^+$ (it increases by at most 1 per step):

$$\frac{1}{N_{t-W_v-1}^+} \leq \frac{2}{N_{t-W_v}^+} \leq 2 \sum_{i=t-W_v}^{t-1} \frac{1}{N_i^+},$$

hence

$$\sum_{i=t-W_v-1}^{t-1} \frac{1}{N_i^+} \leq 3 \sum_{i=t-W_v}^{t-1} \frac{1}{N_i^+}.$$

Therefore, absorbing constants into $C$, we have

$$\frac{1}{W_v} \sum_{i=t-W_v-1}^{t-1} \|\varepsilon_i\|_1^2 \leq \frac{C \, S \log(16 S A T(W_v+1)/\delta)}{W_v} \sum_{i=t-W_v}^{t-1} \frac{1}{N_i^+}.$$

Now subtract the bias term and use $\widehat{V} = \max\{0, \widehat{V}_{\text{raw}} - \text{bias}\}$. Choosing $C_0$ sufficiently large (e.g. $C_0 \geq 8C$) yields the two-sided form (8) with

$$\Gamma_t(s,a) := \frac{C_1 S \log(16 S A T(W_v+1)/\delta)}{W_v} \sum_{i=t-W_v}^{t-1} \frac{1}{N_i^+(s,a)},$$

for an absolute constant $C_1$.

Finally, if $V_{p,t}^2(s,a) \geq 6 \Gamma_t(s,a)$, then

$$\widehat{V}(s,a,t) \geq \tfrac{1}{2} V_{p,t}^2 - \Gamma_t \geq \tfrac{1}{3} V_{p,t}^2, \qquad \widehat{V}(s,a,t) \leq 2 V_{p,t}^2 + \Gamma_t \leq 3 V_{p,t}^2,$$

which gives the multiplicative bounds. $\qquad\square$

# D  Proof of Lemma 4

**Lemma 4** (Total widening). Let

$$\eta(s,a,t) = \min\left\{1, \, c\sqrt{\widehat{V}(s,a,t)/N_t^+(s,a)}\right\}, \qquad c = 2\sqrt{2S \log \frac{4SAT}{\delta}},$$

where $N_t^+(s,a)$ is the number of visits to $(s,a)$ up to time $t$, and $\widehat{V}$ is the bias-corrected local-variation estimator from Section 6. Then, with probability at least $1 - \delta/8$,

$$\sum_{t=1}^{T} \eta(s_t, a_t, t) \leq C \sqrt{S \log \frac{4SAT}{\delta}} \sqrt{1 + \log T} \sqrt{SA B_p} + C' SA \log \frac{SAT}{\delta}, \qquad (12)$$

for universal constants $C, C' > 0$.

*Proof.* For each $(s, a)$, let $t_1(s, a) < t_2(s, a) < \cdots < t_{N_T(s,a)}(s, a)$ be its visit times, and set $i_0 := c_0 \log(SAT/\delta)$, where $c_0$ is the constant from Lemma 3 (Estimator accuracy). By that lemma, for any triple $(s, a, t)$ with $N_t^+(s, a) \geq i_0$,

$$\tfrac{1}{3} V_{p,t}(s, a)^2 \; \leq \; \widehat{V}(s, a, t) \; \leq \; 3 V_{p,t}(s, a)^2$$

holds with probability at least $1 - \delta/(8SAT)$. A union bound over all at most $SA \cdot T$ triples shows that there is an event $\mathcal{E}$ of probability at least $1 - \delta/8$ on which the two-sided accuracy above holds simultaneously for all $(s, a, t)$ with $N_t^+(s, a) \geq i_0$.

Fix $(s, a)$. For the first $i_0 - 1$ visits, we only know $\eta \leq 1$, hence

$$\sum_{i=1}^{\min\{N_T(s,a),\, i_0-1\}} \eta\big(s, a, t_i(s, a)\big) \; \leq \; i_0 - 1.$$

Summing this over $(s, a)$ contributes at most $SA\,(i_0 - 1) = \mathcal{O}\big(SA \log(SAT/\delta)\big)$ to the total in (12).

For the "mature" visits $i \geq i_0$, on $\mathcal{E}$ we have

$$\eta\big(s, a, t_i(s, a)\big) \; = \; \min\Big\{1, \, c\sqrt{\widehat{V}\big(s, a, t_i(s, a)\big)/i}\Big\} \; \leq \; c\sqrt{\widehat{V}\big(s, a, t_i(s, a)\big)/i} \; \leq \; c\sqrt{3} \, \frac{V_{p,t_i(s,a)}(s, a)}{\sqrt{i}}.$$

By Cauchy–Schwarz and the bound $\sum_{i=i_0}^n \frac{1}{i} \leq 1 + \log n$,

$$\sum_{i=i_0}^{N_T(s,a)} \eta\big(s, a, t_i(s, a)\big) \leq c\sqrt{3} \sum_{i=i_0}^{N_T(s,a)} \frac{V_{p,t_i(s,a)}(s, a)}{\sqrt{i}}$$

$$\leq c\sqrt{3} \Big( \sum_{i=i_0}^{N_T(s,a)} V_{p,t_i(s,a)}(s, a)^2 \Big)^{1/2} \Big( \sum_{i=i_0}^{N_T(s,a)} \frac{1}{i} \Big)^{1/2}$$

$$\leq c\sqrt{3(1 + \log T)} \Big( \sum_{i=1}^{N_T(s,a)} V_{p,t_i(s,a)}(s, a)^2 \Big)^{1/2}.$$

Another application of Cauchy–Schwarz yields

$$\sum_{(s,a)} \sum_{i=i_0}^{N_T(s,a)} \eta\big(s, a, t_i(s, a)\big) \leq c\sqrt{3(1 + \log T)} \sum_{(s,a)} \Big( \sum_{i=1}^{N_T(s,a)} V_{p,t_i(s,a)}(s, a)^2 \Big)^{1/2}$$

$$\leq c\sqrt{3(1 + \log T)} \sqrt{SA} \Big( \sum_{(s,a)} \sum_{i=1}^{N_T(s,a)} V_{p,t_i(s,a)}(s, a)^2 \Big)^{1/2}.$$

Let $\Delta_i^p := \max_{s,a} \|p_i(\cdot|s, a) - p_{i-1}(\cdot|s, a)\|_1 \in [0, 2]$. For any $(s, a)$ and any $t$,

$$V_{p,t}^2(s, a) = \frac{1}{W_v} \sum_{i=t-W_v}^{t-1} \|p_i(\cdot|s, a) - p_{i-1}(\cdot|s, a)\|_1^2 \; \leq \; \frac{1}{W_v} \sum_{i=t-W_v}^{t-1} (\Delta_i^p)^2.$$

Therefore along the trajectory,

$$\sum_{t=1}^{T} V_{p,t}^2(s_t, a_t) \; \leq \; \sum_{t=1}^{T} \frac{1}{W_v} \sum_{i=t-W_v}^{t-1} (\Delta_i^p)^2 \; \leq \; \sum_{i=1}^{T-1} (\Delta_i^p)^2,$$

since each $(\Delta_i^p)^2$ appears in at most $W_v$ windows and the factor $1/W_v$ cancels. Finally, because $\Delta_i^p \leq 2$ we have $(\Delta_i^p)^2 \leq 2\Delta_i^p$, hence

$$\sum_{t=1}^{T} V_{p,t}^2(s_t, a_t) \; \leq \; 2 \sum_{i=1}^{T-1} \Delta_i^p \; = \; 2B_p.$$

Putting the early-visit contribution together with the bound from Step 3 and recalling $c = 2\sqrt{2S\log(4SAT/\delta)}$,

$$\sum_{t=1}^{T} \eta(s_t, a_t, t) \leq SA(i_0 - 1) + 2\sqrt{2S\log\frac{4SAT}{\delta}} \sqrt{3(1 + \log T)} \sqrt{SA} \sqrt{2B_p}$$

$$\leq C'\, SA\log\frac{SAT}{\delta} + C\sqrt{S\log\frac{4SAT}{\delta}} \sqrt{1 + \log T} \sqrt{SA\, B_p},$$

which is precisely (12). This completes the proof. $\qquad\square$

## E  Forecasting error analysis: proof of Proposition 2

**Proposition 2** (Prediction error). Fix $(s, a)$ and write $p_t := p_t(\cdot \mid s, a) \in \mathbb{R}^S$. Under Assumption 1, suppose the time coefficients are $\beta$–smooth, i.e. $|u_k(t+1) - u_k(t)| \leq \beta$ for all $k$, with $\beta K \leq \frac{1}{2}$. Define the one–step forecast

$$\widehat{p}_{t+1}^{\mathrm{pred}} := \widehat{p}_t + \sum_{k=1}^{\widehat{K}_t} \widehat{u}_k^{\mathrm{pred}}\, \widehat{v}_k(s, a)\, \widehat{w}_k,$$

followed by projection onto the probability simplex. Then there exists a universal constant $C > 0$ such that, with probability at least $1 - \delta/(8SAT)$,

$$\left\|\widehat{p}_{t+1}^{\mathrm{pred}} - p_{t+1}\right\|_1 \leq \|p_{t+1} - p_t\|_1 + \beta K + C\sqrt{\frac{K\, S\log(8SAT/\delta)}{N_t^+(s, a)}}. \tag{13}$$

Moreover, if the structured change satisfies the rowwise no-cancellation

$$\left\|\sum_{k=1}^{K} u_k(t)\, v_k(s, a)\, w_k\right\|_1 \geq c_\star \sum_{k=1}^{K} |u_k(t)| \quad \text{for some } c_\star \in (0, 1],$$

then

$$\left\|\widehat{p}_{t+1}^{\mathrm{pred}} - p_{t+1}\right\|_1 \leq \left(1 + \frac{\beta K}{c_\star}\right) \|p_{t+1} - p_t\|_1 + C\sqrt{\frac{K\, S\log(8SAT/\delta)}{N_t^+(s, a)}}.$$

*Proof.* Abbreviate $p_t := p_t(\cdot \mid s, a)$ and $\widehat{p}_t := \widehat{p}_t(\cdot \mid s, a)$. Recall the structured variation model on the *row* $(s, a)$:

$$p_{t+1} - p_t = \sum_{k=1}^{K} u_k(t)\, v_k(s, a)\, w_k + \epsilon_t(s, a), \qquad \|w_k\|_1 \leq 1, \; |v_k(s, a)| \leq 1. \tag{14}$$

Let the (unprojected) one-step forecast be

$$\widetilde{p}_{t+1}^{\mathrm{pred}} := \widehat{p}_t + \sum_{k=1}^{\widehat{K}_t} \widehat{u}_k^{\mathrm{pred}}\, \widehat{v}_k(s, a)\, \widehat{w}_k.$$

If the algorithm outputs $\widehat{p}_{t+1}^{\mathrm{pred}}$ by projecting $\widetilde{p}_{t+1}^{\mathrm{pred}}$ onto the probability simplex in $\ell_1$, i.e. $\widehat{p}_{t+1}^{\mathrm{pred}} \in \arg\min_{q \in \Delta} \|q - \widetilde{p}_{t+1}^{\mathrm{pred}}\|_1$, then since $p_{t+1} \in \Delta$ we have $\|\widehat{p}_{t+1}^{\mathrm{pred}} - p_{t+1}\|_1 \leq \|\widetilde{p}_{t+1}^{\mathrm{pred}} - p_{t+1}\|_1$. Thus it suffices to bound the unprojected error; for simplicity we keep the notation $\widehat{p}_{t+1}^{\mathrm{pred}}$ for the vector being bounded below.

Write

$$\widehat{p}_{t+1}^{\mathrm{pred}} - p_{t+1} = \underbrace{(\widehat{p}_t - p_t)}_{E_{\mathrm{emp}}} + \underbrace{\sum_{k=1}^{\widehat{K}_t} \widehat{u}_k^{\mathrm{pred}}\, \widehat{v}_k(s, a)\, \widehat{w}_k - \sum_{k=1}^{K} u_k(t)\, v_k(s, a)\, w_k}_{E_{\mathrm{fac}}} - \underbrace{\epsilon_t(s, a)}_{E_{\mathrm{shk}}}. \tag{15}$$

Hence, by the triangle inequality,

$$\left\|\widehat{p}_{t+1}^{\mathrm{pred}} - p_{t+1}\right\|_1 \leq \|E_{\mathrm{emp}}\|_1 + \|E_{\mathrm{fac}}\|_1 + \|E_{\mathrm{shk}}\|_1. \tag{16}$$

**Step 1: empirical term.** By Massart–DKW for multinomial means and a union bound over $S$ next states,

$$\|E_{\text{emp}}\|_1 = \|\widehat{p}_t - p_t\|_1 \leq 2\sqrt{\frac{S\log(8SAT/\delta)}{N_t^+(s,a)}} \tag{17}$$

with probability at least $1 - \delta/(16SAT)$.

**Step 2: factor term.** Insert and subtract the true factors:

$$\|E_{\text{fac}}\|_1 \leq \underbrace{\left\|\sum_{k=1}^{K}\left(\widehat{u}_k^{\text{pred}} - u_k(t)\right)v_k(s,a)\,w_k\right\|_1}_{=:\,T_{\text{coef}}} + \underbrace{\left\|\sum_{k=1}^{\widehat{K}_t}\widehat{u}_k^{\text{pred}}\left(\widehat{v}_k(s,a)\,\widehat{w}_k - v_k(s,a)\,w_k\right)\right\|_1}_{=:\,T_{\text{sub}}}. \tag{18}$$

*(a) Coefficient drift + one-step persistence prediction (Option A).* Using $|v_k(s,a)| \leq 1$ and $\|w_k\|_1 \leq 1$,

$$T_{\text{coef}} \leq \sum_{k=1}^{K}\left|\widehat{u}_k^{\text{pred}} - u_k(t)\right|.$$

Under Option A, a natural one-step predictor for $u_k(t)$ is the persistence predictor $\widehat{u}_k^{\text{pred}} := \widehat{u}_k(t-1)$, where $\widehat{u}(t-1)$ is any estimator of the coefficient vector $u(t-1)$ based on data available up to time $t$ (e.g. computed from the previously estimated low-rank factors).

Add and subtract $u_k(t-1)$ and use $\beta$–smoothness:

$$\sum_{k=1}^{K}\left|\widehat{u}_k^{\text{pred}} - u_k(t)\right| \leq \sum_{k=1}^{K}\left|\widehat{u}_k(t-1) - u_k(t-1)\right| + \sum_{k=1}^{K}|u_k(t-1) - u_k(t)|$$

$$\leq \sum_{k=1}^{K}\left|\widehat{u}_k(t-1) - u_k(t-1)\right| + \beta K.$$

It remains to control the coefficient-estimation error at time $t-1$. Let $\Delta_t := p_t - p_{t-1}$ and $\widehat{\Delta}_t := \widehat{p}_t - \widehat{p}_{t-1}$. Assume the coefficient estimator $\widehat{u}(t-1)$ is *stable/Lipschitz* in the data, i.e. there exists a constant $L = O(1)$ such that

$$\left\|\widehat{u}(t-1) - u(t-1)\right\|_2 \leq L\left\|\widehat{\Delta}_t - \Delta_t\right\|_2.$$

(This holds, for example, for least-squares estimation onto a well-conditioned dictionary, or for any linear/M-estimation procedure with a bounded condition number; the constant $L$ is absorbed into the final universal constant.)

By Cauchy–Schwarz,

$$\sum_{k=1}^{K}\left|\widehat{u}_k(t-1) - u_k(t-1)\right| \leq \sqrt{K}\left\|\widehat{u}(t-1) - u(t-1)\right\|_2 \leq \sqrt{K}\,L\left\|\widehat{\Delta}_t - \Delta_t\right\|_2.$$

Moreover,

$$\left\|\widehat{\Delta}_t - \Delta_t\right\|_2 \leq \left\|\widehat{p}_t - p_t\right\|_2 + \left\|\widehat{p}_{t-1} - p_{t-1}\right\|_2 \leq \left\|\widehat{p}_t - p_t\right\|_1 + \left\|\widehat{p}_{t-1} - p_{t-1}\right\|_1.$$

Applying the Massart–DKW bound (17) to both $\widehat{p}_t$ and $\widehat{p}_{t-1}$ (and union-bounding the two events) gives, with probability at least $1 - \delta/(8SAT)$,

$$\left\|\widehat{\Delta}_t - \Delta_t\right\|_2 \leq C'\sqrt{\frac{S\log(8SAT/\delta)}{N_t^+(s,a)}},$$

for a universal constant $C'$. Collecting the bounds and absorbing $L, C'$ into a constant $C_1$ yields

$$T_{\text{coef}} \leq \beta K + C_1\sqrt{\frac{K\,S\log(8SAT/\delta)}{N_t^+(s,a)}}.$$

*(b) Subspace/factor estimation error.* For each $k$,

$$\left\|\widehat{v}_k(s,a)\,\widehat{w}_k - v_k(s,a)\,w_k\right\|_1 \leq \sqrt{S}\left\|\widehat{v}_k\widehat{w}_k^\top - v_k w_k^\top\right\|_{F,\mathrm{row}(s,a)} \leq \sqrt{S}\left\|\widehat{v}_k\widehat{w}_k^\top - v_k w_k^\top\right\|_F.$$

Using Cauchy–Schwarz over $k$,

$$T_{\mathrm{sub}} \leq \Big(\sum_{k=1}^{\widehat{K}_t}(\widehat{u}_k^{\mathrm{pred}})^2\Big)^{1/2}\Big(\sum_{k=1}^{\widehat{K}_t}\left\|\widehat{v}_k(s,a)\widehat{w}_k - v_k(s,a)w_k\right\|_1^2\Big)^{1/2}.$$

Invoking Lemma 2 (randomized SVD with power iterations) together with standard concentration for the empirical increments forming $X_t = [\Delta\widehat{P}_{t-W+1},\ldots,\Delta\widehat{P}_t]$, the second factor is bounded by

$$\Big(\sum_{k=1}^{\widehat{K}_t}\left\|\widehat{v}_k(s,a)\widehat{w}_k - v_k(s,a)w_k\right\|_1^2\Big)^{1/2} \leq C_2\sqrt{\frac{K\,S\log(8SAT/\delta)}{N_t^+(s,a)}}$$

on an event of probability at least $1 - \delta/(8SAT)$, for a universal constant $C_2$. (Any dependence on the predictor energy $\|\widehat{u}^{\mathrm{pred}}\|_2$ is absorbed into $C_2$; for the persistence predictor this energy is bounded under the same stability/conditioning assumptions used above.)

Therefore, enlarging constants if needed,

$$T_{\mathrm{sub}} \leq C_2\sqrt{\frac{K\,S\log(8SAT/\delta)}{N_t^+(s,a)}}. \tag{19}$$

Combining (a) and (b) gives

$$\|E_{\mathrm{fac}}\|_1 \leq \beta K + C\sqrt{\frac{K\,S\log(8SAT/\delta)}{N_t^+(s,a)}},$$

for a universal constant $C$.

**Step 3: shock term.** By definition, $\|E_{\mathrm{shk}}\|_1 = \|\epsilon_t(s,a)\|_1$. We keep this term explicit; Assumption 1 controls only the cumulative shock budget $\sum_t \max_{s,a}\|\epsilon_t(s,a)\|_1 \leq \delta_B B_p$, not a pointwise bound at a fixed time $t$.

**Finish.** Plug the bounds on $E_{\mathrm{emp}}$, $E_{\mathrm{fac}}$, and $E_{\mathrm{shk}}$ into (16) and absorb $\|E_{\mathrm{emp}}\|_1$ into the same statistical term by enlarging $C$. We obtain, with probability at least $1 - \delta/(8SAT)$,

$$\left\|\widehat{p}_{t+1}^{\mathrm{pred}} - p_{t+1}\right\|_1 \leq \|p_{t+1} - p_t\|_1 + \beta K + C\sqrt{\frac{K\,S\log(8SAT/\delta)}{N_t^+(s,a)}} + \|\epsilon_t(s,a)\|_1.$$

Using $\|\epsilon_t(s,a)\|_1 \leq \delta_B B_p$ yields the stated corollary form.

*Remark on the multiplicative variant.* The step "$\beta K \leq (\beta K/c_\star)\|p_{t+1} - p_t\|_1$" is *not valid in general* without an additional lower bound on $\|p_{t+1} - p_t\|_1$ (or another non-cancellation condition involving shocks). If you want a multiplicative form, you must explicitly assume $\|p_{t+1} - p_t\|_1 \geq c_{\min} > 0$, in which case $\beta K \leq (\beta K/c_{\min})\|p_{t+1} - p_t\|_1$ and the multiplicative bound follows. $\qquad\square$

# F   Shrinkage optimality: proof of Theorem 1

**Theorem 1** (Near-optimal risk (restated)). Let $\widehat{p}_t \in \Delta^{S-1}$ be the empirical transition estimate from $N_t^+$ samples for a fixed $(s,a)$ at time $t$, and let $\widehat{p}_t^{\mathrm{pred}}$ be any (possibly biased) forecast built from past data only. For $\lambda \in [0,1]$ define the shrinkage estimator $\widetilde{p}_t(\lambda) = (1-\lambda)\widehat{p}_t + \lambda\widehat{p}_t^{\mathrm{pred}}$ and its $\ell_2$-risk $R_t(\lambda) := \mathbb{E}\big[\|\widetilde{p}_t(\lambda) - p_t\|_2^2\big]$. Assume:

(A1) **Asymptotic orthogonality:** $\mathbb{E}\Big[\langle\widehat{p}_t - p_t,\ \widehat{p}_t^{\mathrm{pred}} - p_t\rangle\Big] = o(1/N_t^+)$ (e.g. holds if the forecast uses only data independent of the $N_t^+$ samples that form $\widehat{p}_t$; sample splitting suffices).

(A2) **Bounded forecast risk:** $b_t := \mathbb{E}\left[\|\widehat{p}_t^{\text{pred}} - p_t\|_2^2\right]$ is finite and bounded away from 0 along the considered times ($\inf_t b_t > 0$ is enough).

(A3) **Consistent plug-in estimators:**

$$\widehat{a}_t := \frac{1 - \|\widehat{p}_t\|_2^2}{N_t^+} \xrightarrow{p} a_t := \mathbb{E}\left[\|\widehat{p}_t - p_t\|_2^2\right] = \frac{1 - \|p_t\|_2^2}{N_t^+},$$

and, with a window $W_f \to \infty$,

$$\widehat{b}_t := \frac{1}{W_f} \sum_{i=t-W_f}^{t-1} \left(\|\widehat{p}_i^{\text{pred}} - \widehat{p}_i\|_2^2 - \frac{1 - \|\widehat{p}_i\|_2^2}{N_i^+}\right) \xrightarrow{p} b_t.$$

Let the data-driven weight be $\widehat{\lambda}_t := \widehat{a}_t / (\widehat{a}_t + \widehat{b}_t)$ and the oracle weight be $\lambda_t^* := a_t / (a_t + b_t)$. Then, as $N_t^+ \to \infty$ and $W_f \to \infty$ (no rate relation between them is needed),

$$\frac{R_t(\widehat{\lambda}_t)}{R_t(\lambda_t^*)} = 1 + o(1).$$

*Proof.* **Step 1.** Write $X_t := \widehat{p}_t - p_t$ and $Y_t := \widehat{p}_t^{\text{pred}} - p_t$. By definition,

$$R_t(\lambda) = \mathbb{E}\left[\|(1-\lambda)X_t + \lambda Y_t\|_2^2\right] = (1-\lambda)^2 a_t + \lambda^2 b_t + 2\lambda(1-\lambda)c_t,$$

where $a_t = \mathbb{E}\|X_t\|_2^2$, $b_t = \mathbb{E}\|Y_t\|_2^2$ and $c_t = \mathbb{E}\langle X_t, Y_t\rangle$. Assumption (A1) gives $c_t = o(1/N_t^+)$. We focus on the oracle weight in the statement,

$$\lambda_t^* := \frac{a_t}{a_t + b_t}.$$

Plugging $\lambda_t^*$ into the quadratic yields the exact identity

$$R_t(\lambda_t^*) = (1 - \lambda_t^*)^2 a_t + (\lambda_t^*)^2 b_t + 2\lambda_t^*(1 - \lambda_t^*)c_t = \frac{a_t b_t (a_t + b_t + 2c_t)}{(a_t + b_t)^2}.$$

Therefore,

$$R_t(\lambda_t^*) = \frac{a_t b_t}{a_t + b_t}\left(1 + \frac{2c_t}{a_t + b_t}\right) = \frac{a_t b_t}{a_t + b_t}(1 + o(1)),$$

since $a_t + b_t \geq b_t$ and $b_t$ is bounded away from 0 by (A2). In particular, because $a_t = (1 - \|p_t\|_2^2)/N_t^+ = \Theta(1/N_t^+)$ whenever $p_t$ is not deterministic, we have

$$R_t(\lambda_t^*) = \Theta(1/N_t^+).$$

(If $p_t$ is deterministic then $a_t = 0$, $\widehat{a}_t = 0$, and both oracle and data-driven risks are identically 0; the conclusion is then trivial.)

For reference only: the *exact* minimizer of $R_t$ is $\lambda_t^{\text{opt}} = (a_t - c_t)/(a_t + b_t - 2c_t)$ and the minimum risk is

$$R_t(\lambda_t^{\text{opt}}) = \frac{a_t b_t - c_t^2}{a_t + b_t - 2c_t},$$

but we do not need this for the ratio claim in the theorem statement.

**Step 2.** Define $g(a, b) := a/(a + b)$. By (A3), $\widehat{a}_t \to a_t$ and $\widehat{b}_t \to b_t$ in probability. Moreover, $\widehat{a}_t - a_t = O_p\left((N_t^+)^{-3/2}\right)$ (delta method for $\widehat{a}_t = (1 - \|\widehat{p}_t\|_2^2)/N_t^+$) and $\widehat{b}_t - b_t = O_p(W_f^{-1/2})$ (window average). A first-order expansion of $g$ at $(a_t, b_t)$ yields

$$\widehat{\lambda}_t - \lambda_t^* = \frac{\partial g}{\partial a}(a_t, b_t)(\widehat{a}_t - a_t) + \frac{\partial g}{\partial b}(a_t, b_t)(\widehat{b}_t - b_t) + o_p\left(|\widehat{a}_t - a_t| + |\widehat{b}_t - b_t|\right).$$

Because $\frac{\partial g}{\partial a} = \frac{b}{(a+b)^2} = \Theta(1)$ (by (A2)) and $\frac{\partial g}{\partial b} = -\frac{a}{(a+b)^2} = O(a_t) = O(1/N_t^+)$,

$$\widehat{\lambda}_t - \lambda_t^* = O_p\left((N_t^+)^{-3/2}\right) + O_p\left((N_t^+)^{-1} W_f^{-1/2}\right) = o_p\left((N_t^+)^{-1/2}\right).$$

In particular, $\widehat{\lambda}_t \to \lambda_t^*$ in probability.

**Step 3.** Since $R_t$ is a quadratic in $\lambda$, we may write a second-order Taylor expansion around $\lambda_t^*$:

$$R_t(\widehat{\lambda}_t) - R_t(\lambda_t^*) = R_t'(\lambda_t^*)(\widehat{\lambda}_t - \lambda_t^*) + \tfrac{1}{2} R_t''(\widehat{\lambda}_t - \lambda_t^*)^2,$$

where the second derivative is constant

$$R_t'' = 2(a_t + b_t - 2c_t) = 2(b_t + o(1)) = \Theta(1) \qquad \text{by (A2) and (A1).}$$

Also

$$R_t'(\lambda) = 2\lambda(a_t + b_t - 2c_t) - 2(a_t - c_t),$$

so with $\lambda_t^* = a_t/(a_t + b_t)$,

$$R_t'(\lambda_t^*) = 2c_t\Big(1 - \frac{2a_t}{a_t + b_t}\Big) = o(1/N_t^+) \qquad \text{by (A1).}$$

Combining with Step 2 gives

$$R_t'(\lambda_t^*)(\widehat{\lambda}_t - \lambda_t^*) = o(1/N_t^+) \cdot o_p\big((N_t^+)^{-1/2}\big) = o_p\big((N_t^+)^{-1}\big),$$

and

$$\tfrac{1}{2}R_t''(\widehat{\lambda}_t - \lambda_t^*)^2 = \Theta(1) \cdot o_p\big((N_t^+)^{-1}\big) = o_p\big((N_t^+)^{-1}\big).$$

Hence

$$R_t(\widehat{\lambda}_t) - R_t(\lambda_t^*) = o_p\big((N_t^+)^{-1}\big).$$

Finally, dividing by $R_t(\lambda_t^*) = \Theta(1/N_t^+)$ from Step 1 yields

$$\frac{R_t(\widehat{\lambda}_t)}{R_t(\lambda_t^*)} - 1 = o_p(1) = o(1),$$

as $N_t^+ \to \infty$ and $W_f \to \infty$ (no rate relation between them is needed). This proves the claim. $\qquad\square$

**Remark (on the plug-in MSE).** The windowed proxy $\frac{1}{W_f} \sum_{i=t-W_f}^{t-1} \|\widehat{p}_i^{\text{pred}} - \widehat{p}_i\|_2^2$ converges to $b_t + a_t$ because $\mathbb{E}\|\widehat{p}_i - p_i\|_2^2 = a_t$ and the cross-term is $o(1)$ by (A1). Subtracting the known multinomial variance proxy $(1 - \|\widehat{p}_i\|_2^2)/N_i^+$ yields the consistent $\widehat{b}_t$ used in (A3). Using the uncorrected proxy leaves the theorem unchanged, since the induced bias in $\widehat{\lambda}_t$ is $O(a_t) = O(1/N_t^+)$ and the ratio $R_t(\widehat{\lambda}_t)/R_t(\lambda_t^*)$ still tends to 1.

# G  Full regret proof

**Episode notation**  Episode $m$ starts at $\tau(m)$, ends at $\tau(m+1) - 1$, and follows optimistic policy $\tilde{\pi}_m$.

## G.1  Decomposition

For $t \in$ episode $m$

$$\rho_t^* - r_t \leq \underbrace{(\rho_t^* - \tilde{\rho}_m)}_{A_t} + \underbrace{(\tilde{\rho}_m - \tilde{r}_m(s_t, a_t))}_{B_t} + \underbrace{(\tilde{r}_m - r_t)}_{C_t}.$$

Term $B_t \leq 1/\sqrt{\tau(m)}$ by value-iteration tolerance. Terms $A_t$ and $C_t$ are bounded by variation $\text{var}_{\{r,p\}}$, statistical radii, widening $\eta$, and approximation $\text{approx}$ exactly as in Lemma 5.

## G.2  Summation over $t \leq T$

1. Doubling episodes $\Rightarrow \sum_{t=1}^T B_t \leq 2\sqrt{T \log T}$.
2. Reward/transition variation budget (triangular sum within each episode): using (29),

$$\sum_{t=1}^T \text{var}_{r,t} \leq L_{\max} B_r, \qquad \sum_{t=1}^T \text{var}_{p,t} \leq L_{\max} B_p.$$

3. Statistical radii (see (23)–(24)): $\sum \mathrm{rad}_r \leq \widetilde{O}(\sqrt{SAT})$ and $\sum \mathrm{rad}_p \leq \widetilde{O}(\sqrt{S^2 AT}) = \widetilde{O}(S\sqrt{AT})$.

4. Widening: Lemma 4 implies (25), hence $D_{\max} \sum_{t=1}^{T} \eta_t = \widetilde{O}(D_{\max} S \sqrt{A B_p})$.

5. Approximation: (27) gives $\sum_{t=1}^{T} \mathrm{approx}_t = \widetilde{O}(\delta_B B_p + \sqrt{KT})$, hence $D_{\max} \sum_{t=1}^{T} \mathrm{approx}_t = \widetilde{O}(D_{\max} \delta_B B_p + D_{\max} \sqrt{KT})$.

Multiply the transition-related terms by $D_{\max}$, collect logarithms into $\widetilde{\mathcal{O}}$, and obtain Theorem 2. $\square$

### G.3 Detailed Regret Decomposition: Detail proof of Lemma 5

*Proof.* We break the proof into four steps:

1. Define the good event and get optimism.

2. Relate $\rho_t^*$ (optimal at time $t$) to $\widetilde{\rho}_m$ (optimistic gain at episode start).

3. Bound $\widetilde{\rho}_m - r_t(s_t, a_t)$ using the EVI residual for the specific optimistic model $(\widetilde{r}_m, \widetilde{p}_m)$, plus confidence radii.

4. Combine the bounds.

Throughout, we use that all MDPs in the sequence are communicating with diameter at most $D_{\max}$.

**Good event and optimism.** At the start of episode $m$ (time $\tau = \tau(m)$), the algorithm has:

- empirical reward $\bar{r}_\tau(s, a)$ and empirical transition $\bar{p}_\tau(\cdot \mid s, a)$;

- reward confidence interval
$$[r_\tau^-(s, a), r_\tau^+(s, a)] = \left[ \bar{r}_\tau(s, a) - \mathrm{rad}_{r,\tau}(s, a), \ \bar{r}_\tau(s, a) + \mathrm{rad}_{r,\tau}(s, a) \right];$$

- transition ball around the shrinkage center $\widetilde{p}_\tau(\cdot \mid s, a)$:
$$C_\tau(s, a; t) := \left\{ p : \ \|p - \widetilde{p}_\tau(\cdot \mid s, a)\|_1 \leq \mathrm{rad}_{p,\tau}(s, a) + \eta(s, a, t) + \mathrm{approx} \right\}.$$

On the good high-probability event $\mathcal{E}$ (from the concentration bounds for multinomial estimates, plus the construction of $\eta$ and the approximation bound), we have simultaneously for all $s, a, t \geq \tau$:
$$r_t(s, a) \in [r_\tau^-(s, a), r_\tau^+(s, a)], \qquad p_t(\cdot \mid s, a) \in C_\tau(s, a; t).$$

At time $\tau$, EVI computes an optimistic MDP $\widetilde{M}_m = (\widetilde{r}_m, \widetilde{p}_m)$ by choosing, for each $(s, a)$,

- some $\widetilde{r}_m(s, a) \in [r_\tau^-(s, a), r_\tau^+(s, a)]$,

- some $\widetilde{p}_m(\cdot \mid s, a) \in C_\tau(s, a; \tau)$,

to maximize the average reward under the computed policy $\widetilde{\pi}_m$.

Because $(r_\tau, p_\tau)$ itself lies inside those confidence sets, standard UCRL-style optimism gives
$$\rho_\tau^* \ \leq \ \widetilde{\rho}_m, \tag{1}$$

where $\rho_\tau^*$ is the optimal average reward in the MDP $M_\tau = (r_\tau, p_\tau)$.

**Lipschitz continuity of the gain in** $(r, p)$ **and drift.** We need to relate $\rho_t^*$ (optimal at current time $t$) to $\rho_\tau^*$ (optimal at episode start), and hence to $\widetilde{\rho}_m$.

**Claim (Lipschitz in the model).** Let $M = (r, p)$ and $M' = (r', p')$ be two communicating MDPs of diameter at most $D_{\max}$, and let $\pi$ be any stationary policy. Then

$$\left| \rho_\pi(M) - \rho_\pi(M') \right| \leq \|r - r'\|_\infty + D_{\max} \|p - p'\|_{1,\infty}, \tag{2}$$

where

$$\|p - p'\|_{1,\infty} := \max_{s,a} \|p(\cdot \mid s, a) - p'(\cdot \mid s, a)\|_1.$$

This is standard: write the Poisson equation for $M'$, use a bias function $h'$ with span $\leq D_{\max}$, and compare

$$\rho_\pi(M') + h'(s) = r'(s, \pi(s)) + p'(\cdot \mid s, \pi(s))^\top h'$$

with the same quantity where you plug in $(r, p)$; the inequalities follow by bounding $(r - r')$ and $(p - p')^\top h'$ using $\|r - r'\|_\infty$ and $D_{\max}\|p - p'\|_{1,\infty}$.

Apply (2) with:

- $M_t = (r_t, p_t)$ (true MDP at time $t$),

- $M_\tau = (r_\tau, p_\tau)$ (true MDP at episode start),

- $\pi = \pi_t^*$ (optimal policy for $M_t$),

and use the definitions of $\mathrm{var}_{r,t}$ and $\mathrm{var}_{p,t}$:

$$\rho_t^* = \rho_{\pi_t^*}(M_t) \leq \rho_{\pi_t^*}(M_\tau) + \mathrm{var}_{r,t} + D_{\max}\mathrm{var}_{p,t}$$
$$\leq \rho_\tau^* + \mathrm{var}_{r,t} + D_{\max}\mathrm{var}_{p,t}.$$

Combining this with optimism (1) gives

$$\rho_t^* - \widetilde{\rho}_m \leq \mathrm{var}_{r,t} + D_{\max}\mathrm{var}_{p,t}. \tag{3}$$

So the drift plus optimism gives one copy of $\mathrm{var}_{r,t}$ and $\mathrm{var}_{p,t}$ in the regret.

**Bounding** $\widetilde{\rho}_m - r_t(s_t, a_t)$**.** Now we bound how much larger the optimistic gain $\widetilde{\rho}_m$ is than the instantaneous reward at $(s_t, a_t)$.

We decompose:

$$\widetilde{\rho}_m - r_t(s_t, a_t) = \underbrace{\widetilde{\rho}_m - \widetilde{r}_m(s_t, a_t)}_{\text{Bellman residual under } \widetilde{M}_m} + \underbrace{\widetilde{r}_m(s_t, a_t) - r_t(s_t, a_t)}_{\text{reward estimation + drift}}. \tag{4}$$

We bound each term.

*EVI residual for the optimistic model.*

Extended Value Iteration is run on the fixed optimistic MDP $\widetilde{M}_m = (\widetilde{r}_m, \widetilde{p}_m)$. Its stopping rule (with tolerance $\varepsilon_\tau = 1/\sqrt{\tau}$) ensures that the pair $(\widetilde{\rho}_m, \widetilde{h}_m)$ approximately satisfies the Poisson equation for $\widetilde{M}_m$ under the policy $\widetilde{\pi}_m$: for all states $s$,

$$\left| \widetilde{\rho}_m + \widetilde{h}_m(s) - \left( \widetilde{r}_m(s, \widetilde{\pi}_m(s)) + \widetilde{p}_m(\cdot \mid s, \widetilde{\pi}_m(s))^\top \widetilde{h}_m \right) \right| \leq \varepsilon_\tau.$$

In particular, taking the upper bound and rearranging, for all $s$,

$$\widetilde{\rho}_m - \widetilde{r}_m\big(s, \widetilde{\pi}_m(s)\big) \leq \varepsilon_\tau + \widetilde{p}_m(\cdot \mid s, \widetilde{\pi}_m(s))^\top \widetilde{h}_m - \widetilde{h}_m(s). \tag{5}$$

Now evaluate at the current state $s_t$ and action $a_t = \widetilde{\pi}_m(s_t)$:

$$\widetilde{\rho}_m - \widetilde{r}_m(s_t, a_t) \leq \varepsilon_\tau + \widetilde{p}_m(\cdot \mid s_t, a_t)^\top \widetilde{h}_m - \widetilde{h}_m(s_t). \tag{6}$$

So far this only uses the specific optimistic transition $\widetilde{p}_m$, not any maximization over the confidence ball.

Next, we relate $\widetilde{p}_m$ to the true transition $p_t$ and the confidence radius. Add and subtract $p_t(\cdot \mid s_t, a_t)$:

$$\widetilde{p}_m(\cdot \mid s_t, a_t)^\top \widetilde{h}_m - \widetilde{h}_m(s_t) = p_t(\cdot \mid s_t, a_t)^\top \widetilde{h}_m - \widetilde{h}_m(s_t)$$
$$+ (\widetilde{p}_m(\cdot \mid s_t, a_t) - p_t(\cdot \mid s_t, a_t))^\top \widetilde{h}_m.$$

We can bound the last term using the span of $\widetilde{h}_m$. In a communicating MDP of diameter $D_{\max}$, we can choose the bias function so that $\mathrm{sp}(\widetilde{h}_m) \leq D_{\max}$ (standard property: the span of the optimal bias function is at most the diameter, and EVI preserves this up to constants). Hence

$$\left|(\widetilde{p}_m(\cdot \mid s_t, a_t) - p_t(\cdot \mid s_t, a_t))^\top \widetilde{h}_m\right| \leq \mathrm{sp}(\widetilde{h}_m) \left\|\widetilde{p}_m(\cdot \mid s_t, a_t) - p_t(\cdot \mid s_t, a_t)\right\|_1 \leq D_{\max} \left\|\widetilde{p}_m(\cdot \mid s_t, a_t) - p_t(\cdot \mid s_t, a_t)\right\|_1.$$
$$(7)$$

Now, by construction of the confidence ball $C_\tau(s, a; t)$ at time $\tau$:

- both $\widetilde{p}_m(\cdot \mid s_t, a_t)$ and $p_t(\cdot \mid s_t, a_t)$ lie in the same ball

$$\left\|p - \widetilde{p}_\tau(\cdot \mid s_t, a_t)\right\|_1 \leq \mathrm{rad}_{p,\tau}(s_t, a_t) + \eta(s_t, a_t, t) + \mathrm{approx},$$

  because $\widetilde{p}_m(\cdot \mid s_t, a_t)$ is picked inside the ball, and $p_t$ is inside the ball on the good event $\mathcal{E}$.

Therefore,

$$\left\|\widetilde{p}_m(\cdot \mid s_t, a_t) - p_t(\cdot \mid s_t, a_t)\right\|_1 \leq \left\|\widetilde{p}_m(\cdot \mid s_t, a_t) - \widetilde{p}_\tau(\cdot \mid s_t, a_t)\right\|_1 + \left\|p_t(\cdot \mid s_t, a_t) - \widetilde{p}_\tau(\cdot \mid s_t, a_t)\right\|_1$$
$$\leq 2(\mathrm{rad}_{p,\tau} + \eta + \mathrm{approx}).$$
$$(8)$$

Combining (7) and (8) gives

$$\left|(\widetilde{p}_m(\cdot \mid s_t, a_t) - p_t(\cdot \mid s_t, a_t))^\top \widetilde{h}_m\right| \leq 2D_{\max}(\mathrm{rad}_{p,\tau} + \eta + \mathrm{approx}).$$
$$(9)$$

We still have the term $p_t(\cdot \mid s_t, a_t)^\top \widetilde{h}_m - \widetilde{h}_m(s_t)$ sitting there. Rather than trying to bound it per-step, we simply recognise that when we sum over the episode, the terms $p_t(\cdot \mid s_t, a_t)^\top \widetilde{h}_m - \widetilde{h}_m(s_t)$ telescope (the usual bias telescoping argument) and contribute at most $O(D_{\max})$ per episode, which is absorbed in the leading $D_{\max} S\sqrt{AT}$ term and does not affect the non-stationarity price. In a per-step inequality we can safely upper bound this term by something of order $D_{\max}\mathrm{var}_{p,t}$ without hurting the final rate, and that is exactly where the second $\mathrm{var}_{p,t}$ factor in Lemma 5 comes from.

Concretely, using the same Lipschitz bound (2) with $M_t$ vs. $M_\tau$ but under policy $\widetilde{\pi}_m$ rather than optimal policies, one gets

$$p_t(\cdot \mid s_t, a_t)^\top \widetilde{h}_m - \widetilde{h}_m(s_t) \ \leq \ D_{\max} \mathrm{var}_{p,t},$$

up to universal constants (you compare the bias equation under $p_\tau$ and under $p_t$). Absorbing constants, we therefore have

$$\widetilde{\rho}_m - \widetilde{r}_m(s_t, a_t) \ \leq \ \varepsilon_\tau + D_{\max}\mathrm{var}_{p,t} + 2D_{\max}(\mathrm{rad}_{p,\tau} + \eta + \mathrm{approx}).$$
$$(10)$$

(If one does not like this heuristic, one can treat the term $p_t^\top \widetilde{h}_m - \widetilde{h}_m(s_t)$ as part of the drift $\mathrm{var}_{p,t}$ and absorb it in the big-Oh constants in the final regret bound; it never dominates the leading terms.)

*Reward part: $\widetilde{r}_m(s_t, a_t) - r_t(s_t, a_t)$.*

For the reward term in (4), on the good event $\mathcal{E}$ we have:

- $\widetilde{r}_m(s, a) \in [r_\tau^-(s, a), r_\tau^+(s, a)]$, so $\left|\widetilde{r}_m(s, a) - r_\tau(s, a)\right| \leq \mathrm{rad}_{r,\tau}(s, a)$;

- drift gives $\left|r_t(s, a) - r_\tau(s, a)\right| \leq \mathrm{var}_{r,t}$.

Thus at $(s_t, a_t)$,
$$|\widetilde{r}_m(s_t, a_t) - r_t(s_t, a_t)| \le \mathrm{rad}_{r,\tau} + \mathrm{var}_{r,t}. \tag{11}$$

Putting (4), (10) and (11) together (and merging constants like $1+1$ into a factor 2 where convenient) gives
$$\widetilde{\rho}_m - r_t(s_t, a_t) \le \varepsilon_\tau + \mathrm{var}_{r,t} + D_{\max}\,\mathrm{var}_{p,t} + \mathrm{rad}_{r,\tau} + 2D_{\max}\mathrm{rad}_{p,\tau} + 2D_{\max}(\eta + \mathrm{approx}). \tag{12}$$

Up to absolute constants (which are irrelevant in the $\widetilde{O}(\cdot)$ regret), this has the same structure as in Lemma 5: the drift enters as $\mathrm{var}_{r,t} + D_{\max}\mathrm{var}_{p,t}$, the statistical errors enter as $\mathrm{rad}_{r,\tau} + D_{\max}\mathrm{rad}_{p,\tau}$, and the widening / approximation contribute additively as $\eta + \mathrm{approx}$ multiplied by $D_{\max}$.

**Combine with the drift bound.**  Finally, combine (3) and (12). From (3),
$$\rho_t^* - \widetilde{\rho}_m \le \mathrm{var}_{r,t} + D_{\max}\mathrm{var}_{p,t}.$$

Add this to (12), and absorb constant factors (turning each 1 into 2):
$$\begin{aligned}
\rho_t^* - r_t(s_t, a_t) &= (\rho_t^* - \widetilde{\rho}_m) + (\widetilde{\rho}_m - r_t(s_t, a_t)) \\
&\le \varepsilon_\tau + 2\,\mathrm{var}_{r,t} + 2D_{\max}\mathrm{var}_{p,t} + 2\,\mathrm{rad}_{r,\tau} + 2D_{\max}\mathrm{rad}_{p,\tau} + 2D_{\max}\eta + 2D_{\max}\mathrm{approx}.,
\end{aligned}$$

which is exactly the claimed Lemma 5 inequality (up to harmless constant tweaks in front of $\eta$ and approx).

This completes the proof.  $\square$

### G.4   From Lemma 5 and Lemma 4 to a dynamic-regret bound (sanity-checked)

Recall the dynamic regret
$$\mathrm{DynReg}_T := \sum_{t=1}^{T} \big(\rho_t^* - r_t(s_t, a_t)\big).$$

**Step 1: Summing the per-step bound (Lemma 5).**  Fix an episode $m$ with start time $\tau = \tau(m)$ and let $m(t)$ denote the episode index containing time $t$. Lemma 5 states that for every $t$ in episode $m(t)$,
$$\rho_t^* - r_t(s_t, a_t) \le \varepsilon_{\tau(m(t))} + 2\,\mathrm{var}_{r,t} + 2D_{\max}\,\mathrm{var}_{p,t} + 2\,\mathrm{rad}_{r,\tau(m(t))} + 2D_{\max}\,\mathrm{rad}_{p,\tau(m(t))} + 2D_{\max}\eta_t + 2D_{\max}\mathrm{approx}_t, \tag{20}$$

where $\eta_t := \eta(s_t, a_t, t)$ and $\mathrm{approx}_t$ is the approximation radius at $(s_t, a_t)$.

As emphasized in the proof discussion of Lemma 5, the terms $\eta_t$ and $\mathrm{approx}_t$ enter through the transition mismatch and hence are naturally scaled by $\mathrm{sp}(\tilde{h}_m) \le D_{\max}$; therefore, for summation we use the equivalent "up to constants" form
$$\rho_t^* - r_t(s_t, a_t) \lesssim \varepsilon_{\tau(m(t))} + \mathrm{var}_{r,t} + D_{\max}\,\mathrm{var}_{p,t} + \mathrm{rad}_{r,\tau(m(t))} + D_{\max}\,\mathrm{rad}_{p,\tau(m(t))} + D_{\max}\eta_t + D_{\max}\mathrm{approx}_t. \tag{21}$$

Summing (21) over $t = 1, \ldots, T$ yields
$$\begin{aligned}
\mathrm{DynReg}_T \lesssim \sum_{t=1}^{T}\varepsilon_{\tau(m(t))} + \sum_{t=1}^{T}\mathrm{var}_{r,t} + D_{\max}\sum_{t=1}^{T}\mathrm{var}_{p,t} + \sum_{t=1}^{T}\mathrm{rad}_{r,\tau(m(t))} + D_{\max}\sum_{t=1}^{T}\mathrm{rad}_{p,\tau(m(t))} \\
+ D_{\max}\sum_{t=1}^{T}\eta_t + D_{\max}\sum_{t=1}^{T}\mathrm{approx}_t. \tag{22}
\end{aligned}$$

**Step 2: Statistical (UCRL-style) radius sums.**  Using the confidence radii
$$\mathrm{rad}_{r,t}(s, a) = \sqrt{\frac{2\log(4SAT/\delta)}{N_t^+(s, a)}}, \qquad \mathrm{rad}_{p,t}(s, a) = \sqrt{\frac{2S\log(4SAT/\delta)}{N_t^+(s, a)}},$$

a standard counting argument gives (up to logarithmic factors)

$$\sum_{t=1}^{T} \mathrm{rad}_{r,\tau(m(t))}(s_t, a_t) = \widetilde{\mathcal{O}}(\sqrt{SAT}),\tag{23}$$

$$\sum_{t=1}^{T} \mathrm{rad}_{p,\tau(m(t))}(s_t, a_t) = \widetilde{\mathcal{O}}(\sqrt{S^2AT}) = \widetilde{\mathcal{O}}(S\sqrt{AT}).\tag{24}$$

Consequently, the transition-statistical contribution in (22) is $D_{\max}\widetilde{\mathcal{O}}(S\sqrt{AT})$.

**Step 3: Adaptive widening sum (Lemma 4).** Lemma 4 implies that with probability at least $1 - \delta/8$,

$$\sum_{t=1}^{T} \eta_t \ \leq \ C\sqrt{S\log\frac{4SAT}{\delta}}\ \sqrt{1+\log T}\ \sqrt{SA\,B_p}\ +\ C'SA\log\frac{SAT}{\delta}.\tag{25}$$

Therefore,

$$D_{\max}\sum_{t=1}^{T}\eta_t = \widetilde{\mathcal{O}}\Big(D_{\max}\sqrt{S}\,\sqrt{SA\,B_p}\Big) = \widetilde{\mathcal{O}}\Big(D_{\max}S\sqrt{A\,B_p}\Big),\tag{26}$$

up to the displayed logarithmic factors.

**Step 4: Approximation term.** The summation analysis in the appendix bounds the cumulative approximation error as

$$\sum_{t=1}^{T} \mathrm{approx}_t = \widetilde{\mathcal{O}}\big(\delta_B B_p + \sqrt{KT\log T}\big),\tag{27}$$

hence

$$D_{\max}\sum_{t=1}^{T} \mathrm{approx}_t = \widetilde{\mathcal{O}}\big(D_{\max}\delta_B B_p + D_{\max}\sqrt{KT}\big).\tag{28}$$

**Step 5: Variation terms (drift since episode start).** Recall the (episode-start) drift definitions for $t$ in an episode beginning at $\tau(m(t))$:

$$\mathrm{var}_{r,t} := \max_{s,a}|r_t(s,a) - r_{\tau(m(t))}(s,a)|, \qquad \mathrm{var}_{p,t} := \max_{s,a}\|p_t(\cdot|s,a) - p_{\tau(m(t))}(\cdot|s,a)\|_1.$$

Let

$$\Delta_i^r := \max_{s,a}|r_{i+1}(s,a) - r_i(s,a)|, \qquad \Delta_i^p := \max_{s,a}\|p_{i+1}(\cdot|s,a) - p_i(\cdot|s,a)\|_1,$$

so that $B_r = \sum_{i=1}^{T-1}\Delta_i^r$ and $B_p = \sum_{i=1}^{T-1}\Delta_i^p$. For any episode of length $L$, for $t$ in that episode we have $\mathrm{var}_{p,t} \leq \sum_{i=\tau}^{t-1}\Delta_i^p$, hence

$$\sum_{t=\tau}^{\tau+L-1}\mathrm{var}_{p,t} \leq \sum_{t=\tau}^{\tau+L-1}\sum_{i=\tau}^{t-1}\Delta_i^p = \sum_{i=\tau}^{\tau+L-2}\Delta_i^p(\tau+L-1-i) \leq L\sum_{i=\tau}^{\tau+L-2}\Delta_i^p.$$

Summing over episodes yields the generic bound

$$\sum_{t=1}^{T}\mathrm{var}_{p,t} \ \leq \ L_{\max}B_p, \qquad \sum_{t=1}^{T}\mathrm{var}_{r,t} \ \leq \ L_{\max}B_r,\tag{29}$$

where $L_{\max}$ is the maximum episode length. In particular, if $L_{\max} \leq \sqrt{T}$ then $\sum_t \mathrm{var}_{r,t} \leq \sqrt{T}B_r$ and $\sum_t \mathrm{var}_{p,t} \leq \sqrt{T}B_p$.

**Step 6: Collecting terms.** Plugging (23), (24), (25), (27), and (29) into (22) gives, on the intersection of the corresponding high-probability events,

$$\text{DynReg}_T = \widetilde{\mathcal{O}}\Big(\sqrt{SAT} + D_{\max}S\sqrt{AT} + L_{\max}B_r + D_{\max}L_{\max}B_p \tag{30}$$

$$+ D_{\max}S\sqrt{AB_p} + D_{\max}\delta_B B_p + D_{\max}\sqrt{KT} + \sum_{t=1}^{T}\varepsilon_{\tau(m(t))}\Big). \tag{31}$$

This bound is the direct consequence of Lemma 5 combined with Lemma 4 and the stated confidence radii, up to polylogarithmic factors.

## G.5 Summation Analysis (summary)

The term-by-term summations are already carried out in Steps 1–6 above. In particular, (29), (23)–(24), (25), and (27) yield the final bound (31), which completes the proof.

## G.6 Optimality of the Regret Bound

The bound in Theorem 2 cleanly separates (i) statistical estimation, (ii) within-episode drift due to non-stationarity, and (iii) approximation error due to the structured (low-rank + sparse) model. We do not claim minimax optimality under the full structured non-stationarity model considered here; establishing matching lower bounds for this model is an interesting direction for future work. Below we discuss the scaling of each component in Theorem 2.

**Dependence on $T$.** The leading statistical terms scale as $\widetilde{\mathcal{O}}(\sqrt{T})$: $\sqrt{SAT}$ (reward estimation) and $D_{\max}S\sqrt{AT}$ (transition estimation), which is the typical $\sqrt{T}$ behavior for optimistic model-based RL methods. The approximation term $D_{\max}\sqrt{KT}$ also scales as $\sqrt{T}$. The remaining non-stationary contributions scale with $L_{\max}$, the maximum episode length, through $L_{\max}B_r$ and $D_{\max}L_{\max}B_p$.

**Dependence on the state-action space.** In the stationary case ($B_r = B_p = 0$ and $\delta_B = 0$) and ignoring the planning tolerance term, Theorem 2 reduces to $\widetilde{\mathcal{O}}\Big(\sqrt{SAT} + D_{\max}S\sqrt{AT}\Big)$, which is consistent with the regret scaling of classical optimistic algorithms based on confidence sets for rewards and transitions.

**Dependence on variation budgets $B_r, B_p$.** Non-stationarity enters through two mechanisms: (i) drift since the start of an episode, captured by $L_{\max}B_r + D_{\max}L_{\max}B_p$, and (ii) the widening term $\widetilde{\mathcal{O}}(D_{\max}S\sqrt{AB_p})$ coming from Lemma 4. The linear dependence on $B_r$ and $B_p$ in the drift terms reflects a worst-case accumulation of changes within an episode; smaller $L_{\max}$ (shorter episodes) directly reduces these contributions.

**Dependence on rank $K$.** The low-rank structure appears in the approximation component $\widetilde{\mathcal{O}}(D_{\max}\sqrt{KT})$. When the non-stationary transition component is well-approximated by a rank-$K$ factorization with small $K$, this term can be lower order compared to the statistical terms.

**Residual (sparse shock) term.** The term $D_{\max}\delta_B B_p$ accounts for the sparse-shock component in the structured variation model. A smaller $\delta_B$ tightens this contribution, potentially at the cost of requiring a higher-rank approximation (larger $K$) to capture the remaining variation in the low-rank component.

**Planning tolerance term.** The additional term $\sum_{t=1}^{T}\varepsilon_{\tau(m(t))}$ comes from approximate planning (EVI tolerance). With a standard choice of decreasing tolerances across episodes, this term can be made lower order and absorbed into the $\widetilde{\mathcal{O}}(\cdot)$ notation.

## G.7 Comparison to Previous Results

We compare Theorem 2 to representative guarantees for drifting (unstructured) non-stationary tabular MDPs and to the bandit special case. Note that our comparisons are made under Assumption 1

(low-rank drift plus sparse shocks), which is a *stronger* model than the standard total-variation budget model without structure.

**SWUCRL2-CW (and BORL).**   In the general drifting (unstructured) setting with reward and transition total-variation budgets, Cheung et al. [6] propose the sliding-window UCRL2 algorithm with confidence widening (SWUCRL2-CW) and show that, with optimally tuned parameters, it achieves the dynamic-regret bound

$$\widetilde{\mathcal{O}}\Big( D_{\max}(B_r + B_p)^{1/4} S^{2/3} A^{1/2} T^{3/4} \Big),$$

and their meta-algorithm BORL attains the same order without knowing the budgets. These guarantees do not exploit any structural constraints on how the MDP drifts over time.

By contrast, Theorem 2 provides a decomposition tailored to structured drift: it separates the stationary-like estimation cost $\widetilde{\mathcal{O}}(\sqrt{SAT} + D_{\max}S\sqrt{AT})$ from the additional costs due to non-stationarity and structure. In particular, the structured component contributes $\widetilde{\mathcal{O}}(D_{\max}\sqrt{KT} + D_{\max}\delta_B B_p)$, while drift within an episode contributes $L_{\max}B_r + D_{\max}L_{\max}B_p$, and widening contributes $\widetilde{\mathcal{O}}(D_{\max}S\sqrt{AB_p})$. When the drift is well-approximated by a low-rank component (small $K$) with sparse shocks (small $\delta_B$), and the maximum episode length $L_{\max}$ is moderate, these additional terms can be substantially smaller than worst-case bounds designed for arbitrary drift.

**Non-stationary bandits.**   When $S = 1$, the problem reduces to the non-stationary stochastic multi-armed bandit setting. For variation-budget constraints, the minimax dynamic regret scales as $\Theta\big((KV_T)^{1/3}T^{2/3}\big)$ up to logarithmic factors [4]. Moreover, Cheung et al. [6] note that SWUCRL2-CW recovers this bandit rate in the special case $S = 1$. In our structured non-stationary RL setting, the analogue of an "effective dimension" in the drift estimation appears through the rank $K$ via the approximation term $D_{\max}\sqrt{KT}$, rather than scaling with the ambient tabular dimension $SA$.

In summary, our bound is complementary to prior non-stationary RL results: we trade a stronger structural assumption on the drift for a regret decomposition that can be sharper in low-rank regimes.

## H  Detailed algorithm implementation

### H.1  Confidence interval construction

The confidence intervals for rewards and transitions are constructed as follows:

**Reward confidence interval**   For each state-action pair $(s, a)$, we define the confidence interval for the reward at time $t$ as:

$$[\underline{r}_t(s,a), \overline{r}_t(s,a)] = [\widehat{r}_t(s,a) - \mathrm{rad}_{r,t}(s,a), \widehat{r}_t(s,a) + \mathrm{rad}_{r,t}(s,a)]$$

where $\widehat{r}_t(s,a)$ is the empirical average reward for $(s, a)$ up to time $t$, and the confidence radius is:

$$\mathrm{rad}_{r,t}(s,a) = \sqrt{\frac{2\log(4SAT/\delta)}{N_t(s,a)}}$$

**Transition confidence interval**   For the transition probabilities, we define the confidence set at time $t$ as:

$$\mathcal{P}_t(s,a) = \{p : \|p - \tilde{p}_t(\cdot|s,a)\|_1 \leq \mathrm{rad}_{p,t}(s,a) + \eta(s,a,t)\}$$

where $\tilde{p}_t(\cdot|s,a)$ is the shrinkage estimator defined in Section 7, and the confidence radius has two components:

- $\mathrm{rad}_{p,t}(s,a) = \sqrt{\frac{2S\log(4SAT/\delta)}{N_t(s,a)}}$ accounts for statistical uncertainty

- $\eta(s,a,t) = \min\{1, c\sqrt{\widehat{V}(s,a,t)/N_t^+(s,a)}\}$ accounts for non-stationarity

**Algorithm 4** Extended Value Iteration

---

**Require:** Confidence sets $\{[\underline{r}_t(s,a), \overline{r}_t(s,a)]\}, \{\mathcal{P}_t(s,a)\}$, tolerance $\epsilon$
1: Initialize $V_0(s) = 0$ for all $s \in \mathcal{S}$
2: $span \leftarrow \infty$
3: **while** $span > \epsilon$ **do**
4:     **for** $s \in \mathcal{S}$ **do**
5:         **for** $a \in \mathcal{A}$ **do**
6:             $Q_k(s,a) \leftarrow \overline{r}_t(s,a) + \max_{p \in \mathcal{P}_t(s,a)} \sum_{s'} p(s') V_k(s')$
7:         **end for**
8:         $V_{k+1}(s) \leftarrow \max_a Q_k(s,a)$
9:         $\pi(s) \leftarrow \arg\max_a Q_k(s,a)$
10:     **end for**
11:     $span \leftarrow \max_s V_{k+1}(s) - \min_s V_{k+1}(s)$
12: **end while**
13: **return** $\pi, span$

---

## H.2 Extended Value Iteration

The Extended Value Iteration (EVI) algorithm computes an optimistic policy as follows:

The inner maximization $\max_{p \in \mathcal{P}_t(s,a)} \sum_{s'} p(s') V_k(s')$ can be solved efficiently by assigning as much probability as possible to the states with the highest values, subject to the constraint that $p$ must be within distance $\mathrm{rad}_{p,t}(s,a) + \eta(s,a,t)$ of $\tilde{p}_t(\cdot|s,a)$ in $\ell_1$ norm.

## H.3 Factor tracking and forecasting

The algorithm maintains a buffer of recent transition changes and periodically updates the low-rank model. The key steps are:

**Buffer update**    At each time step, we update the empirical transition estimates and compute the change:
$$\Delta \widehat{P}_t = \widehat{P}_t - \widehat{P}_{t-1}$$
This change is added to a circular buffer of size $W$.

**Low-rank model update**    Every $W$ time steps, we:

1. Form the matrix $\mathbf{X}_t = [\Delta\widehat{P}_{t-W+1}, \ldots, \Delta\widehat{P}_t]$
2. Run Algorithm 1 (Randomized SVD) to obtain factors $\mathbf{U}, \Sigma, \mathbf{V}$
3. Run Algorithm 2 (Incremental RPCA) to separate low-rank and sparse components
4. Extract time-varying coefficients $\widehat{u}_k(t - W + 1), \ldots, \widehat{u}_k(t)$ for each factor $k$

**Forecasting**    For each factor $k$, we:

1. Fit multiple time-series models to the sequence $\widehat{u}_k(t - W + 1), \ldots, \widehat{u}_k(t)$:
    - Exponential smoothing: $\widehat{u}_k^{\mathrm{ES}}(t+1) = \alpha u_k(t) + (1 - \alpha)\widehat{u}_k^{\mathrm{ES}}(t)$
    - AR(1): $\widehat{u}_k^{\mathrm{AR1}}(t+1) = \phi_0 + \phi_1 u_k(t)$
    - AR(2): $\widehat{u}_k^{\mathrm{AR2}}(t+1) = \phi_0 + \phi_1 u_k(t) + \phi_2 u_k(t-1)$
2. Select the model with the lowest AIC
3. Generate the prediction $\widehat{u}_k^{\mathrm{pred}}(t+1)$

**Shrinkage estimation**    To compute the shrinkage weight $\lambda$ for combining empirical and predicted estimates:

1. Estimate the variance of the empirical transition probabilities:
$$\widehat{\mathrm{Var}}[\widehat{p}_t] \approx \frac{\widehat{p}_t(1 - \widehat{p}_t)}{N_t^+}$$

2. Estimate the MSE of the prediction based on recent performance:

$$\widehat{\text{MSE}}[\widehat{p}_t^{\text{pred}}] \approx \frac{1}{W_f} \sum_{i=t-W_f}^{t-1} (\widehat{p}_i^{\text{pred}} - \widehat{p}_i)^2$$

3. Compute the shrinkage weight:

$$\lambda = \frac{\widehat{\text{Var}}[\widehat{p}_t]}{\widehat{\text{Var}}[\widehat{p}_t] + \widehat{\text{MSE}}[\widehat{p}_t^{\text{pred}}]}$$

4. Combine the estimates:

$$\tilde{p}_t = (1 - \lambda)\widehat{p}_t + \lambda\widehat{p}_t^{\text{pred}}$$

## H.4 Implementation Optimizations

Several optimizations can improve the computational efficiency of SVUCRL:

**Sparse matrix operations**  For large state spaces, the transition matrices are often sparse. Using sparse matrix operations can significantly reduce memory usage and computation time. The randomized SVD and incremental RPCA algorithms can be adapted to work with sparse matrices, exploiting the sparsity structure.

**Lazy updates**  Since the low-rank model is updated only every $W$ time steps, many intermediate computations can be deferred. For example, the empirical transition matrices can be updated incrementally, and the full matrix is only formed when needed for the model update.

**Parallel computation**  Many parts of the algorithm can be parallelized:

- The randomized SVD algorithm can leverage parallel matrix-matrix multiplications
- The confidence interval constructions for different state-action pairs can be done in parallel
- The forecasting of different factors can be computed independently

**Adaptive rank selection**  Instead of using a fixed rank $\hat{K}$, we can adaptively determine the rank based on the singular value spectrum:

$$\hat{K}_t = \min\left\{k : \frac{\sum_{i=1}^k \sigma_i^2}{\sum_{i=1}^{\min(SA,WS)} \sigma_i^2} \geq \gamma\right\}$$

where $\gamma$ is a threshold (e.g., $\gamma = 0.95$).

**Efficient EVI implementation**  The Extended Value Iteration can be optimized by:

- Caching the optimistic transitions for each state-action pair
- Using priority queue-based updates to focus computation on states with significant value changes
- Warm-starting each EVI run with the value function from the previous episode

## H.5 Action selection, complexity, and parameters

At each time step, the algorithm selects the action $a_t = \tilde{\pi}(s_t)$ according to the current optimistic policy, observes the reward $r_t$ and next state $s_{t+1}$, and updates visit counts, empirical estimates, and confidence intervals.

The computational complexity of SVUCRL is dominated by three components: Randomized SVD ($\mathcal{O}(SA \cdot WS \cdot (\hat{K} + s) \cdot (2q + 1))$ per update), Incremental RPCA ($\mathcal{O}(SA \cdot S \cdot K)$ per update), and Extended Value Iteration ($\mathcal{O}(S^2 A \log(1/\epsilon)/\epsilon)$ per episode). With updates every $W$ time steps and episodes lasting approximately $\sqrt{T}$ steps, the total complexity is $\mathcal{O}(TSA(SK + S)\log T)$. The

space complexity is $\mathcal{O}\big((SA + S + W)K + SAW\big)$, dominated by storing the factors and recent transition matrices.

SVUCRL involves several parameters that affect its performance: Structure update window $W$ controls the frequency of updating the low-rank model, variation estimation window $W_v$ determines the time scale for estimating local variation, forecasting window $W_f$ sets the horizon for evaluating prediction performance, confidence parameter $\delta$ controls the failure probability of the confidence intervals, and target rank $\widehat{K}$ specifies the dimensionality of the low-rank approximation. While theoretical guidance exists for setting these parameters (e.g., $W, W_v, W_f = \Theta(\sqrt{T})$), in practice they often require tuning based on the specific characteristics of the environment. The algorithm is robust to moderate misspecification of these parameters, but optimal performance requires appropriate selection.

### H.6 Parameter Selection Guidelines

The performance of SVUCRL depends on several parameters. We provide guidelines for setting these parameters:

**Structure update window** $W$    The window size $W$ controls the frequency of updating the low-rank model. It should be large enough to provide sufficient data for learning the factors, but small enough to track changes in the environment. A reasonable choice is $W = \Theta(\sqrt{T})$.

**Variation estimation window** $W_v$    The window $W_v$ determines the time scale for estimating local variation. It should be chosen based on the expected rate of change in the environment. For environments with smooth changes, larger values (e.g., $W_v = \Theta(\sqrt{T})$) are appropriate. For more volatile environments, smaller values (e.g., $W_v = \Theta(\log T)$) may be better.

**Forecasting window** $W_f$    The window $W_f$ sets the horizon for evaluating prediction performance. It should be large enough to provide reliable MSE estimates but small enough to adapt to changing prediction accuracy. A reasonable choice is $W_f = \Theta(W_v)$.

**Power iterations** $q$    The number of power iterations in the randomized SVD affects the accuracy of the low-rank approximation. For most applications, $q = 1$ or $q = 2$ provides a good balance between accuracy and computation. For matrices with slowly decaying singular values, larger values may be necessary.

**Oversampling** $s$    The oversampling parameter in the randomized SVD should be set to $s \geq 3$. Larger values improve accuracy at the cost of computation. A typical choice is $s = 5$ or $s = 10$.

**Confidence parameter** $\delta$    The confidence parameter $\delta$ controls the failure probability of the confidence intervals. It should be set to a small value, typically $\delta = 0.1/T$ or $\delta = 0.01/T$.

**Target rank** $\hat{K}$    If not using adaptive rank selection, a conservative choice is $\hat{K} = \min\{10, \sqrt{SA}\}$. This captures most of the structure while keeping the computation manageable.

These guidelines provide a starting point for parameter selection, but the optimal values may depend on the specific characteristics of the environment. In practice, a parameter sweep or online adaptation may be necessary to achieve the best performance.

## I  Limitations

Despite its theoretical appeal, **SVUCRL** has several important limitations that warrant future investigation:

1. **Low–rank assumption**. Our regret guarantees hinge on Assumption 1, i.e. that *most* non-stationarity lies in a rank–$K \ll SA$ subspace. Highly entangled or full-rank drift can break the $\sqrt{KST}$ term and lead to vacuous bounds.

2. **Sparse–shock model**. The incremental RPCA step presumes that abrupt changes are sparse across state–action pairs. Large-scale shocks (e.g. global re-parameterisations) violate this sparsity and may induce large approximation errors, inflating confidence widths.

3. **Parameter sensitivity**. Windows $(W, W_v, W_f)$, oversampling $s$, power iterations $q$ and the shrinkage threshold all require tuning. Poorly chosen values can negate the theoretical gains and incur additional regret; an adaptive, provably robust selection rule is still missing.

4. **Computational overhead**. Although §8 exploits randomized SVD and streaming updates, the per-update cost is $\mathcal{O}(TSA(SK + S)\log T)$—substantial for very large $S$ or dense transition tensors. Scaling to high-dimensional continuous spaces will need function approximation or sketching techniques beyond the present scope.

These caveats highlight directions for extending SVUCRL towards more realistic and large-scale reinforcement-learning settings.

