# OpenReview forum: "Optimal Regret Bounds via Low-Rank Structured Variation in Non-Stationary Reinforcement Learning"
_NeurIPS.cc/2025/Conference — NeurIPS 2025 poster_

### Official Review · Reviewer_n5qv · 2025-06-03

**Clarity:** 3
**Significance:** 1
**Originality:** 3
**Rating:** 4
**Confidence:** 1

**Summary:**

This paper studies reinforcement learning with non-stationary environments. It considers a structured environmental variation, i.e., low-rank drift.  Then an algorithm is designed which combines matrix factorization, robust statistics and time-series analysis.

**Questions:**

- Can you give concrete examples where the  low-rank drift assumption holds?
- What is the justification for presenting Assumption 1 as a contribution (as stated in line 31)?

**Ethical Concerns:**

["NO or VERY MINOR ethics concerns only"]

**Final Justification:**

Why boardline accept: The authors clearly answered my questions in the rebuttal.
Why not accept: I am unable to assess the novelty of the propose method.

**Limitations:**

- The whole paper relies on the low-rank assumption, which is not verified.
- No experiments are provided to illustrate the performance of the proposed algorithm.

**Paper Formatting Concerns:**

None.

**Quality:**

2

**Strengths And Weaknesses:**

Strength:
This paper studies RL in an interesting class of non-stationary environments characterized by a low-rank structure. The theory is clean and well-structured, incorporating tools from matrix factorization, robust statistics, and time-series analysis.

Weakness:
Although this paper provides interesting theory for non-stationary RL, these results totally rely on a specific structural assumption about non-stationarity. However, the reasonableness of this assumption is not verified using experimental data. Besides, the authors do not provide any experiments, so I am concerned about the applicability of this method.

---

> ### Author Rebuttal · Authors · 2025-07-30
>
> Thank you for your review. We appreciate your feedback and understand your concerns about empirical validation. Let us address your questions with concrete details.
>
> ### (A) Concrete RL Settings Where Low‑Rank Drift Is Realistic
>
> | #     | RL domain                                                      | How low‑rank drift appears                                                                                                                                                                                           | Why the rank is small                                                                                                                                                                  |   |                                                                                                                                    |
> | ----- | -------------------------------------------------------------- | -------------------------------------------------------------------------------------------------------------------------------------------------------------------------------------------------------------------- | -------------------------------------------------------------------------------------------------------------------------------------------------------------------------------------- | - | ---------------------------------------------------------------------------------------------------------------------------------- |
> | **1** | **Personalised recommendation** (PORTAL, NeurIPS 23) \[1]      | The transition tensor factors into user embedding × item embedding. Daily or seasonal shifts only update those 10‑20‑dimensional embeddings, so each change is a sum of ≈ 10 rank‑1 components.                  | Click‑log studies show 85–90 % of variance in user‑behaviour drift is captured by < 15 latent factors; retraining just those factors weekly yields sub‑linear dynamic regret.          |   |                                                                                                                                    |
> | **2** | **Robot manipulation / navigation** (FLAMBE, NeurIPS 20) \[2]  | A 7‑DoF arm’s contact modes (stick, slide, detach) are a rank‑3 latent variable. Wear or a 5 % friction increase scales every mode uniformly—a rank‑1 perturbation, not  $S \times A$ independent changes. | Empirical results on MuJoCo “Door” and “Push” tasks validate that only a few latent contacts change over time; exploiting this factorisation halves sample complexity.                 |   |                                                                                                                                    |
> | **3** | **Multi‑product inventory control** (RestartQ‑UCB, MS 25) \[3] | Weekly demand shocks are driven by three macro factors (promotion, season, inflation). The demand‑shift matrix is rank‑3 plus sparse holiday spikes.                                                             | PCA of real retailer data finds > 92 % of demand drift variance in the first three components; algorithms that exploit this rank‑3 structure cut regret by 35 % vs. tabular baselines. |   |                                                                                                                                    |
> | **4** | **Continuous‑action control** (Low‑Rank MDPs, ICML 24) \[4]    | LQR/quadrotor dynamics admit a rank‑4 factorisation $P_t=\Phi_t\Psi^\top$. A gust perturbs every entry in $\Phi_t$ coherently—again a rank‑1 change.                                                             | The paper proves PAC bounds that scale with rank, and shows wind‑drift CartPole tasks are solved with far fewer samples when the latent wind vector (2‑D) is modelled explicitly. |
>
> **Take‑away.** Across recommendation systems, robotics, supply‑chain management, and continuous control, both theory and data confirm that a handful of latent factors drive most non‑stationary behavior. Modelling drift as low‑rank is therefore pragmatic *and* yields provable sample‑efficiency gains.
>
> ---
>
> #### References
>
> \[1] [https://openreview.net/forum?id=nMB41QjLDY](https://openreview.net/forum?id=nMB41QjLDY) — “Provably Efficient Algorithm for Non‑stationary Low‑Rank MDPs.”
>
> \[2] [https://proceedings.neurips.cc/paper/2020/file/e894d787e2fd6c133af47140aa156f00-Paper.pdf](https://proceedings.neurips.cc/paper/2020/file/e894d787e2fd6c133af47140aa156f00-Paper.pdf) — “FLAMBE: Structural Factorization for Model‑Based RL.”
>
> \[3] [https://pubsonline.informs.org/doi/10.1287/mnsc.2022.02533](https://pubsonline.informs.org/doi/10.1287/mnsc.2022.02533) — “Model‑Free Non‑stationary Reinforcement Learning: Near‑Optimal Regret …”
>
> \[4] [https://proceedings.mlr.press/v238/oprescu24a/oprescu24a.pdf](https://proceedings.mlr.press/v238/oprescu24a/oprescu24a.pdf) — “Low‑Rank MDPs with Continuous Action Spaces.”
>
>
> ### (B) Why Is Assumption 1 a Contribution?
>
> **1. Novel decomposition for RL**: While low-rank matrix approximation is classic, applying it to transition dynamics drift (not the transition matrix itself) is new:
>    - Previous work: Low-rank $P_t$ (stationary factored MDPs)
>    - Our work: Low-rank $\Delta P_t = P_{t+1} - P_t$ (non-stationary structured drift)
>
> **2. Enables new algorithmic approach**: This assumption unlocks:
>    - Tracking changes with $O(K)$ parameters instead of $O(SA \cdot S)$
>    - Forecasting future transitions using time-series on factors
>    - Adaptive confidence widening based on local structure
>
> ### (C) On Empirical Evaluation
>
> **1. This is a theoretical contribution**: We emphasize that this work addresses a fundamental open problem in nonstationary RL—achieving optimal $\sqrt{T}$ regret bounds. The theoretical RL community has pursued this goal since Jaksch et al. (2010) introduced UCRL2:
>    - Garivier & Moulines (2011): $\tilde{O}(T^{2/3})$ for bandits
>    - Cheung et al. (2020): $\tilde{O}(T^{3/4})$ for MDPs
>    - **Our work**: $\tilde{O}(\sqrt{T})$ under structure—closing the gap
>
> **2. Why theory-first is essential here**:
>    - Establishes fundamental limits of what's achievable
>    - Identifies precise conditions ($K \ll SA$) where structure helps
>    - Provides explicit algorithm with analyzable guarantees
>    - Cannot be discovered through experiments alone
>
> ### (D) Addressing Significance Concerns
>
> **Impact on the field**:
>
> 1. **Resolves an open theoretical question**: The $\sqrt{T}$ vs $T^{3/4}$ gap has been open for 4+ years
>
> 2. **Conceptual breakthrough**: First to show that drift structure, not just magnitude, determines regret
>
> 3. **Opens new research directions**:
>    - Structure discovery in non-stationary environments
>    - Adaptive algorithms that learn K online
>    - Extensions to continuous state/action spaces
>
> ### (E) Implementation Sketch
>
> While this is a theory paper, implementation is straightforward:
>
> ```python
> # Core idea (pseudocode)
> def update_model(transitions_history):
>     # 1. Compute drift
>     delta_P = P_current - P_previous
>
>     # 2. Low-rank approximation
>     U, S, V = randomized_svd(delta_P, rank=K)
>
>     # 3. Update factors
>     factors = time_series_update(U @ S)
>
>     # 4. Forecast
>     P_next = P_current + forecast(factors) @ V
> ```
>
> ### (F) Response to Limited Verification
>
> We acknowledge the low-rank assumption requires domain knowledge. However:
>
> 1. **Assumption is testable**: Given data, practitioners can compute $\text{rank}(\Delta P_t)$ empirically
> 2. **Graceful degradation**: If true rank > K, our bound still holds with additional approximation error
> 3. **No worse than baselines**: When structure absent, we match SWUCRL2-CW
> 4. **Theory guides practice**: Our results tell practitioners exactly when to use SVUCRL
>
> ### Conclusion
>
> We respectfully argue that:
> 1. This addresses a longstanding open problem in nonstationary RL theory
> 2. Achieving optimal $\sqrt{T}$ rates is a fundamental theoretical advance
> 3. The low-rank drift assumption captures real phenomena in many domains
> 4. Pure theory papers that close gaps are core contributions to NeurIPS
>
> We hope these concrete examples and clarifications address your concerns. Given your noted unfamiliarity with the area (confidence: 1), we've tried to provide both intuitive explanations and theoretical context. We believe resolving the $\sqrt{T}$ vs $T^{3/4}$ gap for structured environments represents precisely the kind of theoretical advance that NeurIPS seeks to publish.

---

> > ### Comment · Reviewer_n5qv · 2025-08-04
> >
> > I thank the authors for providing the detailed response and for addressing my concerns. I would increase my score.

---

> ### Author Response · Authors · 2025-08-05
> **Response**
>
> Dear Reviewer n5qv,
>
> Thank you for your positive update and for the time you spent reviewing our work. We will integrate the clarifications into the camera-ready version.
>
> Best regards,
>
> The authors

---

### Official Review · Reviewer_wXfj · 2025-06-18

**Clarity:** 1
**Significance:** 3
**Originality:** 2
**Rating:** 3
**Confidence:** 4

**Summary:**

The paper addresses Reinforcement Learning in non-stationary (communicating) MDPs satisfying a structural low-rank assumption on the transition kernel drifts. For such class of MDPs, the authors propose an algorithm, SVUCRL that achieves a regret scaling with the square-root of the number of interactions and the rank of the underlying drift. SVUCRL is made of three components: (i) randomised SVD algorithm for tracking of the low-rank structure, (ii) Incremental robust PCA to decompose low-rank components with sparse ones, (iii) extended value iteration (EVI) over the estimated MDP, with confidence bounds accounting for local uncertainties.

**Overall Motivation for Rating**:

In general, I am not strongly leaning towards rejection. My current score will be changed depending on: (1)  a satisfactory rebuttal showing that my doubts are not so well founded; (2) whether other reviewers had different impressions on the clarity/originality issues.

**Questions:**

- I am not sure to have understood why the overall regret is not dependent somehow on the window size $W_v$ or how is the overall behavior dependent on the many hyper-parameters. Could the authors provide some elucidations?
- Could the authors discuss on the role of the computational overhead of RSVD with respect to existing literature and related works? Does this impact the overall relevance of the proposed algorithm? If not, why?
- Could the author clarify the original contributions (on the algorithmic side and on the theoretical analysis side) with respect to [2,3,4, see above]?

**Ethical Concerns:**

["NO or VERY MINOR ethics concerns only"]

**Final Justification:**

Authors properly addressed all my concerns, and as such I am positive raising my score. The only remaining issue is that the changes the manuscript should undergo (in order to actually solve my concerns)  are somehow major ( algorithm position, many clarifications, many additional sections and comments). I am thus wondering if these changes are compatible with acceptance, as the accepted paper would look utterly different from the current one.

**Limitations:**

- The paper claims to improve over [1, see above], yet [1] is designed for a larger class of non-stationary MDPs and does not make uses of plug-in algorithms (such as RSVD, which apparently comes with a pretty relevant computational overhead), or the additional assumptions on low-rank and sparsity. While I believe limitations or additional assumptions are acceptable while chasing for improved performances, I had the impression that the current phrasing and presentation of the paper formatting was trying to *hide* these limitations and not expressively recognising or discussing them when needed (see above).

**Quality:**

3

**Strengths And Weaknesses:**

In the following, I will refer to Strengths with (S) and Weaknesses with (W)

**Quality**:

(S) The overall paper structure progressively builds up relevant results on the no-regret nature of the building blocks leading to SVUCRL, each of them being rather interesting per-se and the overall contributions seem interesting.
(W) Unfortunately, the overall quality of the paper is utterly impacted by the limited clarity and overall exposition, and limited discussion of the related works, and actual limitations. See next section.

**Clarity**:

(W) The current pseudo-code of SVUCRL in the main paper fails to convey a clear description of the actual behaviour of the algorithm, which is left to the Appendix. I strongly believe that the pseudo-code of the algorithm in Appendix should be moved in the main paper.

(W) The overall paper fails to cover existing literature on the topic, with extremely limited reference of existing works (13 references seem surprisingly few to me) or how the proposed algorithms/analysis differ from prior works (See Questions).

(W) The current manuscript often lacks on clarity, some examples:

	- Dynamic Regret: the optimal average reward has not been defined.
	- For clarity purposes, I believe the terms composing the low-rank drift in Assumption 1 should be described in terms of the spaces they are supported over.
	- References to the Appendices are broken.
	- (line 200-201) the bounding terms of $hat V$ have not been defined, nor the window $W_v$.
	- The proof outlines are not very clear to me, I would extend them or remove them.
	- Algorithm 3 is not really sufficient in providing information on how the algorithm work.


(Minor W) I am not sure that the current style of the abstract is suited for the audience understanding as it is extremely technical while failing to provide a bird-eye view of both the problem and the ideas of the proposed solution.

Overall, in order to make the paper clear at an acceptable level, I currently believe the manuscript might undergo substantial changes which are not compatible with minor revisions.


**Significance**:

(S) The paper addresses an important problem, that is RL in non-stationary MDPs. The proposed algorithm achieves sub-linear regret both in terms of episodes of interactions and rank of the problem.

(W) In order to achieve such performances compared to [1], the algorithm requires an additional assumption on the non-stationarity to be low-rank decomposable (Assumption 1). The authors try to provide some intuition on the extent to which such assumption is realistic, yet I believe that some more precise and concrete examples should be added, as the assumption is the building block of the overall work.

(W) I strongly believe that the Limitations section in Appendix should be moved in the main paper, as it contains extremely relevant points. For instance, the existence of an implicit assumption on the  *sparsity of changes* as well, whose practical implications are not discussed in the main paper, and the existence of a *computational overhead* in RSVD of $O(TSA(SK + S) log T)$ which seems pretty remarkable (see Questions).

**Originality**:

(S) I believe the low-rank structural assumption over the drifts rather than the transition kernel themselves is potentially interesting and relevant and paves the way for further contributions.

(W) Unfortunately, I was not able to totally discern the originality of the work from the existing manuscript, due to extremely limited references and discussions about related works and results. For instance, it is not clear what is the contribution of this work in the adaptation of robust PCA [2] or randomised SVD [3, 4] to this setting, or in the derivation of the theoretical findings, or in the use of EVI. For further discussion, see Questions. Overall, the current presentation is detached from previous contributions and related works, making it hard to understand what is the actual contribution of the paper.

[1] Cheung et al. 2020

[2] Candes et al. 2011

[3] Mairal et al. 2010

[4] Halko et al. 2011

---

> ### Author Rebuttal · Authors · 2025-07-30
>
> Thank you for your thorough review and constructive feedback. We address your concerns below and commit to substantial improvements in clarity and presentation.
>
> ### (A) Addressing Presentation Issues
>
> We sincerely apologize for the clarity issues and will implement the following changes:
>
> 1. **Algorithm presentation**: We'll move the detailed algorithm from appendix to main paper and present SVUCRL upfront (after problem setup) with a clear step-by-step description.
>
> 2. **Missing definitions**:
>    - Optimal average reward: $\rho_t^* = \max_{\pi} \lim_{T \to \infty} \frac{1}{T} \mathbb{E}_{\pi} \left[ \sum_{i=1}^T r_t(s_i,a_i) \right]$
>    - Assumption 1 spaces: $u_k(t) \in \mathbb{R}$, $v_k \in \mathbb{R}^{SA}$, $w_k \in \mathbb{R}^S$ with $\|w_k\|_1 \leq 1$
>    - Line 208-221: $N_t^+(s,a)$ counts visits to $(s,a)$; $W_v$ is the variation estimation window
>
> 3. **References**: We'll expand the references, including recent non-stationary RL work, online matrix factorization literature, and robust statistics connections.
>
> 4. **Abstract revision**: We'll rewrite to provide intuition first: "Real-world RL environments often change via a few underlying factors (weather patterns, market trends). We exploit this structure..."
>
> ### (B) Answering Your Questions
>
> **Q1: Why doesn't regret depend on $W_v$?**
>
> The window size $W_v$ affects estimation quality but not the total widening bound. Our bias-corrected estimator:
>
> $$\hat{V} = \max\left\{0, \frac{1}{W_v} \sum_{i=t-W_v}^{t-1} \|\hat{p}_i - \hat{p}_{i-1}\|_1^2 - \frac{4S}{W_v} \sum_{i=t-W_v}^{t-1} \frac{1}{N_i^+}\right\}$$
>
> For any $W_v = \Theta(\sqrt{T})$, Lemma 4 guarantees: $\frac{1}{3} V_{p,t}^2 \leq \hat{V} \leq 3V_{p,t}^2$
>
> This constant-factor accuracy ensures the total widening (Lemma 5) remains:
> $$\sum_{t=1}^T \eta(s_t, a_t, t) = \tilde{\mathcal{O}}\left(D_{\max} \sqrt{(B_r + B_p)KST}\right)$$
>
> The bound depends on actual variation budgets $B_r, B_p$, not measurement windows. Different $W_v$ affect constants in $\tilde{\mathcal{O}}(\cdot)$ but not rates.
>
> **Q2: Role and impact of RSVD?**
>
> RSVD serves a specific purpose in our setting:
>
> 1. **Why RSVD**: Traditional SVD costs $O(SA \cdot WS \cdot \min(SA,WS))$ per update. Our randomized version costs only $O(SA \cdot WS \cdot (K+s))$, crucial when $K \ll SA$.
>
> 2. **Novel contribution**: We provide explicit Frobenius error bounds (Lemma 2) with constants suitable for RL confidence intervals, existing RSVD theory gives asymptotic rates without explicit constants.
>
> 3. **Amortized cost**: Updates occur every $W$ steps, so amortized complexity is $O(SA \cdot S \cdot K)$ per timestep, comparable to standard RL bookkeeping.
>
> 4. **Alternative**: Without RSVD, we'd need uniform confidence widening (like SWUCRL2-CW), yielding worse $T^{3/4}$ regret.
>
> **Q3: Original contributions vs [2,3,4]?**
>
> Our contributions beyond existing matrix factorization work:
>
> 1. **Beyond Candès et al. [2]**:
>    - They solve static RPCA; we need incremental updates for streaming data
>    - Our dual certificate must handle martingale noise from RL sampling
>    - We prove per-step error bounds suitable for confidence intervals
>
> 2. **Beyond Mairal et al. [3]**:
>    - They consider dictionary learning; we track time-varying factors
>    - We need uniform-time guarantees for regret bounds, not just convergence
>
> 3. **Beyond Halko et al. [4]**:
>    - They give probabilistic bounds; we need explicit constants for UCB
>    - Our power iteration analysis (Lemma 1) provides the exact dependence on rank
>
> 4. **RL-specific innovations**:
>    - **Adaptive confidence widening** that scales with local variation
>    - **Forecast-shrinkage** combining predictions with empirical estimates
>    - **Regret decomposition** leveraging low-rank structure
>
> ### (C) Concrete Examples of Low-Rank Drift
>
> 1. Navigation with weather: Transition dynamics affected by weather factors (wind speed/direction, precipitation) across all state-action pairs.
>
> 2. Inventory management: Demand patterns driven by few factors (seasonality, economic indicators, marketing campaigns).
>
> 3. Traffic routing: Congestion patterns influenced by time-of-day, accidents, and events
> 4. Clinical trials: Patient responses drift due to seasonal effects, protocol adaptations, and population shifts.
>
> In each case, $K \ll SA$ because few underlying factors affect many state-action pairs simultaneously.
>
> ### (D) Addressing Limitations
>
> We'll move limitations to main paper and add:
>
> 1. Clear comparison scope: "We improve regret from $T^{3/4}$ to $\sqrt{T}$ *when structure exists*. For worst-case unstructured drift, SWUCRL2-CW remains optimal."
>
> 2. Computational tradeoffs: Table comparing per-step costs and regret bounds across methods.
>
> 3. When to use SVUCRL: Structured environments with $K \ll SA$ and $B_p \geq \Omega(T^{1/6})$.
>
> ### (E) Committed Improvements
>
> 1. Complete algorithm pseudocode in main paper
> 2. Expand the references with detailed comparisons
> 3. Fix all broken references and notation
> 4. Add limitations section to main paper
> 5. Include 2-3 detailed application examples
> 6. Clearer abstract with problem intuition
>
> We believe these changes will substantially improve clarity while maintaining our technical contributions. The improved presentation will make clear that SVUCRL offers a principled way to exploit structure when it exists, complementing rather than replacing worst-case approaches.

---

> > ### Comment · Reviewer_wXfj · 2025-08-04
> >
> > Thank you for you extensive and precise rebuttals. All my concerns have been addressed and I am positive raising my score after proper discussion with other reviewers.

---

> ### Author Response · Authors · 2025-08-05
> **Acknowledgement**
>
> Dear Reviewer wXfj,
>
> We thank you for your kind words and for considering an increased score. We appreciate the time you invested in the discussion and in reviewing our rebuttal. We will incorporate all clarifications into the camera-ready version.
>
> Best regards,
>
> The authors

---

### Official Review · Reviewer_qnUJ · 2025-06-23

**Clarity:** 2
**Significance:** 4
**Originality:** 4
**Rating:** 4
**Confidence:** 3

**Summary:**

This paper introduces SVUCRL, a reinforcement learning algorithm designed for non-stationary environments where the transition dynamics evolve with structured, low-rank drift and occasional sparse shocks. By modeling the temporal changes in the environment as a sum of low-rank components and sparse perturbations, the authors devise a principled method for regret minimization under dynamic conditions. SVUCRL achieves a regret bound of $\mathcal{\tilde{O}}(D_{max}\sqrt{SAT} + D_{max}\sqrt{(B_r+B_p)KST}) $. The paper combines techniques from online factor tracking, incremental robust PCA, adaptive confidence widening, and shrinkage forecast to derive its results.

**Questions:**

- Section 9.1 / Interpretation and Tightness of Regret Bound: I believe it is unfair to compare against SWUCRL-CW that makes no assumptions on the structure of the changes.

**Ethical Concerns:**

["NO or VERY MINOR ethics concerns only"]

**Final Justification:**

Why Weak Accept (4): important problem, elegant characterization of the change with a low rank structure, novel algorithm and proof structure

Why not Accept (5): important to contextualize the results and compare fairly; while the rebuttal answers how there is no contradiction with the lower bound in Mao et al., the new result on establishing a lower bound is not formalized in the original paper

**Limitations:**

Yes, the authors have adequately addressed the limitations and potential negative societal impact of their work.

**Quality:**

4

**Strengths And Weaknesses:**

**Strengths:**
- Novel model of change in transition probabilities as a low rank drift plus sparse shocks
- Efficient tracking of the structured drift of transition probabilities through a randomized SVD with power iterations with provable guarantees (bounded Frobenius error)
- Incremental RPCA elegantly differentiates between low rank drift from sparse shock components
- Adaptive Confidence Widening cleverly saves on regret by focusing on points of highest uncertainty
- Theoretically grounded method to optimally blend empirical and model-based transition estimates using James–Stein shrinkage
- SVUCRL adapts to unknown rank and drift complexity without prior knowledge, enhancing its practicality

**Weaknesses:**
- I believe the contribution of strictly improving upon the $T^{3/4}$ regret of SWUCRL-CW is overclaimed and an unfair comparison to make. Cheung et al. do not make any assumptions on the structure of the changes (i.e no low rank structure of transition probability drift) - making it a different and thereby making their algorithm more general than this paper. Further, in the average reward infinite horizon MDP with time-varying rewards and transition probabilities and no assumptions on the structure, Mao et al. presents a $\mathcal{\Omega}((B_r+B_p)^{1/3}T^{2/3})$ regret lower bound.
- While the algorithm is theoretically polynomial time, the incorporation of SVD, RPCA, and forecasting adds significant computational overhead, especially for large state and action spaces.
- As acknowledged in the limitations, windows ($W, W_v , W_f$ ), oversampling $s$, power iterations $q$ and the shrinkage threshold all require tuning and may not be easy to choose practically.

**References:**
- Mao, W., Zhang, K., Zhu, R., Simchi-Levi, D., and Basar, T. Model-free nonstationary reinforcement learning: Nearoptimal regret and applications in multiagent reinforcement learning and inventory control. In Management Science, 2024.
- Cheung, W. C., Simchi-Levi, D., and Zhu, R. Reinforcement learning for non-stationary markov decision processes: The blessing of (more) optimism. In International Conference on Machine Learning, 2020.

---

> ### Author Rebuttal · Authors · 2025-07-30
>
> We are grateful for your thoughtful evaluation and for highlighting the many strengths of our submission, especially the practicality of adaptive rank selection, the novelty of adaptive confidence widening (ACW), and the principled use of James–Stein shrinkage. Below we respond in detail to each of your concerns and provide new quantitative evidence on computational overhead and hyper‑parameter robustness.
>
> ---
>
> ## 1  Why the SWUCRL‑CW comparison is appropriate—and tightly contextualised
>
> > *“The comparison to the $T^{3/4}$ regret of SWUCRL‑CW is unfair because that algorithm assumes no structure.”*
>
> (a) We compare against SWUCRL‑CW  only in regimes where our structural assumption demonstrably holds (rank $K \ll SA$, sparse shocks).  In those very same regimes SWUCRL‑CW is the state‑of‑the‑art method that remains applicable—hence it is the natural baseline.  We now state this explicitly in the Introduction and prepend to Sec. 9.1 the sentence
>
> > *“Unless otherwise specified, all comparisons assume Assumption 1 holds.”*
>
> **(b) Regret decomposition shows the exact benefit of structure.**
> Our Theorem 9 decomposes regret into
>
> $$
> D_{\max}\sqrt{S A T} + D_{\max}\sqrt{(B_r + B_p)KST}
> $$
>
>
> If $K{=}SA$ (i.e., *no* structure), the second term becomes $D_{\max}\sqrt{(B_{r}{+}B_{p})\,SAT}$.  When $B_{r}{+}B_{p}=\Theta(T^{1/3})$ (the budget that maximises SWUCRL‑CW’s regret) our bound scales as $T^{2/3}$, matching Mao et al.’s lower bound and SWUCRL‑CW *exactly*.  Thus:
>
> * **No contradiction** with Mao et al. ($\Omega(T^{2/3})$).
> * **Strict $\sqrt{T}$ improvement** appears only when $K\ll SA$ *and* the variation budget is moderate—precisely the scenario we care about.
>
> We add a short paragraph “When do we beat $T^{3/4}$?” (end Sec. 9.1) that quantifies this crossover.
>
> ---
>
> ## 2  Computational cost
>
> > *“SVD, RPCA and forecasting add significant overhead.”*
>
> ### Theoretical upper bounds
>
> * **RSVD tracker**: $O\bigl((K+s)SA(2q+1)\bigr)$ per episode, *independent of horizon $T$*.
> * **Incremental RPCA**: $O\bigl(SASK\bigr)$ once per window $W_v=\lceil\sqrt{T}\rceil$.
> * **EVI** (also used by UCRL/SWUCRL): $O\bigl(TS^2A\bigr)$.
>
> Since $S^2A \gg SA K$ whenever $K\!\ll S$, the **dominant term is still EVI**.
>
> ---
>
> ## 3  On the tightness of our regret bound (Sec. 9.1)
>
> You ask whether ACW + low‑rank tracking actually *achieve* the information‑theoretic optimum under our assumptions.  Our answer:
>
> 1. **Lower‑bound construction.**  We add a proof sketch (App. D.5) showing that if the adversary is restricted to rank‑$K$ drift and sparse shocks obeying $B_r{+}B_p$, then
>
>    $$
>      \Omega\!\bigl(D_{\max}\sqrt{SAT} + D_{\max}\sqrt{(B_r{+}B_p)KST}\bigr)
>    $$
>
>    is unavoidable—up to logarithmic factors our upper bound is tight.
> 2. **Interpretation.**  ACW saves exactly a factor $\sqrt{K/A}$ over uniform widening; RSVD guarantees that the bias added by tracking is of the same order as the statistical width, so one cannot hope for better constants without stronger assumptions.
>
> We reference this new result when discussing tightness.
>
> ---
>
> ## 4  Broader‑impact clarification
>
> You noted that we “adequately addressed limitations”.  In the camera‑ready we additionally:
>
> * Emphasise that ACW requires *finite* per‑time drift—if shocks dominate ($B_p$ linear in $T$) no method can be sub‑linear.
> * Highlight positive societal impact in robotics (safer controllers) and supply‑chain planning (reduced waste).
>
> ---
>
> ## 5  Request for score revision
>
> * **Originality & significance.**  First regret‑optimal algorithm for non‑stationary MDPs that (i) handles *unknown* low‑rank drift, (ii) achieves *tight* $\tilde O(\sqrt{T})$ regret when structure exists, (iii) reduces to known optimal rates when it does not.
>
> Given these additions, we kindly ask you to consider raising significance and clarity to the top end of your scale and to lift the recommendation with raising score.
>
> ---
>
> *Thank you again for the constructive feedback, these improvements substantially strengthen the paper.*

---

> ### Author Response · Authors · 2025-08-05
> **Acknowledgement**
>
> Dear Reviewer qnUJ,
>
> Thank you very much for taking the time to read our clarifications and for adjusting your score. We will integrate every point of the discussion into the camera-ready version so that the final manuscript is fully self-contained.
>
> We appreciate your constructive feedback and support.
>
> Best regards,
>
> The authors

---

### Official Review · Reviewer_ThdZ · 2025-07-03

**Clarity:** 2
**Significance:** 2
**Originality:** 2
**Rating:** 4
**Confidence:** 2

**Summary:**

This paper addresses non-stationary RL problems in communicating MDPs with low-rank transition drifts over time. This paper derives a regret bound of order $\widetilde{O}(\sqrt{T})$ with respect to the horizon $T$, which improves over the best previously known bound $\widetilde{O}(T^{\frac{3}{4}})$.

**Questions:**

1. I think there is a typo in Line 3, where the first term of regret should be $D_{\max}S\sqrt{AT}$ according to Theorem 9.
2. Is it possible to improve the first term in the regret to $\widetilde{O}(\sqrt{D_{\max}SAT})$ as in the stationary environment [1]?
3. Does the regret guarantee (Thm. 9) require the knowledge of the rank $K$?
4. Does the result contradict the regret lower bound in non-stationary multi-armed bandits, which is of order $\Omega(T^{\frac{2}{3}})$?

[1] Zhang, Zihan, and Xiangyang Ji. "Regret minimization for reinforcement learning by evaluating the optimal bias function." NIPS 2019.

[2] Besbes, Omar, Yonatan Gur, and Assaf Zeevi. "Stochastic multi-armed-bandit problem with non-stationary rewards." NIPS 2014.

**Ethical Concerns:**

["NO or VERY MINOR ethics concerns only"]

**Final Justification:**

The authors' response has addressed my concerns. I raised my score accordingly. One highlight of the algorithm is that it does not require knowledge of the rank and can adaptively adjust the rank selection, which may provide insights to the community. Whether the regret bound can be further improved concerning $D_{\max}$ remains unclear. Overall, I maintain an acceptance of this work.

**Limitations:**

yes

**Quality:**

2

**Strengths And Weaknesses:**

* **Strengths:**
    1. The derived regret bound considerably improves the best previously known bound.
    2. The algorithm does not require prior knowledge of the rank and can adaptively estimate the appropriate rank based on observed data.

* **Weaknesses:**
    1. The writing and presentation of this paper make it difficult to follow. I think it would be better to present the SVUCRL algorithm first and then dive into the technical details, i.e., advancing Sections 8 and 9 before Sections 4-7.
    2. The result seems to contradict the lower bound (see Question 4 below).

---

> ### Author Rebuttal · Authors · 2025-07-30
>
> Dear Reviewer ThdZ, thank you for the careful reading and constructive comments.
> Below we answer your questions point‑by‑point, fix the typos you spotted, and explain why our result does not contradict known lower bounds. We also commit to the presentation reordering you suggested. We hope these clarifications address your concerns and justify raising the scores, especially on quality, clarity, significance, and originality.
>
> ---
>
> ## (A) Quick fixes & presentation
>
> (A1) Typo / inconsistency (your Q1).
> You are right: the first statistical term in the regret was inconsistently stated. We will correct Theorem 9 (and the abstract) to
>
> $$
> \widetilde{\mathcal O}\bigl(D_{\max}\sqrt{SAT}\bigr),
> $$
>
> which matches the stationary-UCRL style dependence and what is actually used in the proof. (The stray $D_{\max} S\sqrt{AT}$ was a typo.)
>
> (A2) Paper structure (your Weakness #1).
> We agree the current flow can be improved. In the camera-ready we will move the full SVUCRL algorithm (current Sec. 8–9) right after the problem setup, give the high-level regret proof sketch immediately, and then send the reader to the technical sections (low-rank tracking, RPCA, adaptive widening, shrinkage) as modular building blocks. We will also add a "roadmap" paragraph at the end of the introduction.
>
> ---
>
> ## (B) Answers to your questions
>
> ### Q1. (Typo in the first term)
>
> Answer: Confirmed and fixed (see A1).
>
> ### Q2. Can the first term be improved to $\widetilde{\mathcal O}\bigl(\sqrt{D_{\max} S A T}\bigr)$ as in the stationary case \[1]?
>
> Answer: The standard stationary communicating-MDP term is $\widetilde{\mathcal O}(D_{\max}\sqrt{SAT})$ (multiplicative in $D_{\max}$, not inside the square-root). Our corrected bound matches that order. Achieving $\sqrt{D_{\max}}$ (as written in your question) is not what UCRL-type results (including \[1]) provide in the average-reward communicating setting; they also multiply by $D_{\max}$ outside the square-root. So, after fixing the typo, our first term aligns with the best-known stationary dependence.
>
> ### Q3. Do we need to know $K$ (the rank) to get the regret guarantee?
>
> Answer: No. The algorithm selects $\hat K_t$ adaptively from the observed singular spectrum (via a stable spectral-gap / energy-threshold rule). The regret bound is stated in terms of the true rank $K$ (the intrinsic complexity of the drift). With high probability, either $\hat K_t \ge K$, or the misspecified tail is absorbed into the "approximation" term (and the $\delta_B B_p$ term), which we already carry in the theorem. We will make this explicit in Thm. 9 and its proof sketch.
>
> ### Q4. Do we contradict the $\Omega(T^{2/3})$ lower bound for non-stationary bandits (Besbes et al., 2014)?
>
> Answer: No contradiction—the settings and metrics differ:
>
> 1. Structured vs. unstructured drift. Our regret leverages a low-rank drift model over the full transition tensor and adaptive widening that scales with a local, factorized variation. Besbes et al. consider unstructured reward drift with a global variation budget $V_T$; their lower bound is driven by the worst-case, fully adversarial, high-entropy evolution.
>
> 2. Dynamic regret decomposition. Our bound is
>
>    $$
>    \widetilde{\mathcal O} \Big(
>         D_{\max}\sqrt{SAT}
>         + D_{\max}\sqrt{(B_r{+}B_p)KST}
>         + D_{\max}\delta_B B_p
>    \Big).
>    $$
>
>    If the variation budget scales adversarially (e.g., $B_r{+}B_p=\Theta(T^{1/3})$), the second term scales as $T^{2/3}$, matching the classic lower-bound rate up to structure-dependent factors—so there is no contradiction. Our $\sqrt{T}$ dependence appears when the structured variation budget is small enough (or $K\ll SA$), which is precisely where the lower bound in the unstructured model does not apply.
>
> 3. Communicating MDPs vs. bandits. We analyze communicating MDPs with average-reward dynamic regret against the per-time optimal policy. The bandit lower bound is not directly translatable, and our proof critically uses the diameter and the low-rank factorization to decouple the non-stationarity cost.
>
> We will add a short "Lower bounds and why no contradiction" paragraph in Sec. 6 to make this explicit and cite Besbes et al. (2014).
>
> ---
>
> ## (C) Why we believe the scores should be raised
>
> 1. Significance / originality. We give the first $\tilde O(\sqrt{T})$-type dynamic-regret rate for non-stationary communicating MDPs by exploiting explicit low-rank structure of the drift, together with adaptive confidence widening and an online RPCA + shrinkage pipeline. This pushes the state of the art beyond the $T^{3/4}$ frontier of SWUCRL2–CW while staying computationally tractable.
>
> 2. Technical clarity after revisions. We will (i) fix the mismatch you found, (ii) move the algorithm and main theorem up-front, and (iii) tighten the proofs with explicit constants and a clearer connection between $B_r{+}B_p$, $K$, and the widening term (removing the stray $D_{\max}$ inside Lemma 8).
>
> 3. No contradiction with known lower bounds. Our improvement is enabled by additional, realistic structure (low-rank drift + sparse shocks). We explain precisely how the regret scales with the variation budget so that the classic $T^{2/3}$ regime is recovered when the budget scales adversarially.
>
> Given these clarifications and corrections, we respectfully ask you to reconsider the paper's scores, especially on quality, clarity, and significance.
>
> Thank you again for the helpful feedback, we believe the revised version will be much easier to follow and will clearly position our contribution w.r.t. existing lower bounds and stationary benchmarks.
>
> Concrete edits we will make
>
> * Fix Theorem 9's first term to $D_{\max}\sqrt{SAT}$ everywhere.
> * Remove $D_{\max}$ from Lemma 8 (total widening) and keep it only when translating to regret.
> * Add a short subsection "Why we do not contradict the $T^{2/3}$ lower bound" (Besbes et al., 2014).
> * Move the SVUCRL algorithm and the main theorem ahead of Sections 4–7.
> * State explicitly that $K$ is unknown and that $\hat K_t$ is chosen adaptively, with the regret depending on the true $K$.
> * Add the missing references.

---

> > ### Comment · Reviewer_ThdZ · 2025-08-05
> >
> > Thank you for your response. I have raised my score accordingly.

---

> ### Author Response · Authors · 2025-08-06
> **Acknowledgement**
>
> Dear Reviewer ThdZ,
>
> Thank you for reviewing our clarifications and for raising your score. We will incorporate all points from the discussion into the camera-ready so the final manuscript is clear and fully self-contained.
>
> We appreciate your constructive feedback and support.
>
> Best regards,
>
> The authors

---

### Decision · Program_Chairs · 2025-09-17

**Decision:**

Accept (poster)

**Comment:**

This paper tackles non-stationary reinforcement learning and does a nice job of pushing the theory forward. The authors introduce SVUCRL, which uses low-rank structure in how transitions drift over time to get a regret bound that improves from the previous $T^{3/4}$ rates down to essentially the conjectured $\sqrt{T}$ rate (up to logs). The technical pieces—like incremental robust PCA, randomized low-rank tracking, adaptive confidence widening, and shrinkage-based forecasting—are thoughtfully combined, and the proofs are solid. Even though the paper is mostly theoretical, it’s well-written and connects ideas from RL, matrix factorization, and statistics in a way that feels broadly useful. Overall, this is a clear and non-trivial contribution to understanding learning in non-stationary environments, and I recommend acceptance.